# Neural Bootstrapper

**Minsuk Shin**[1]*, **Hyungjoo Cho**[2]*, **Hyun-seok Min**[3], **Sungbin Lim**[4]†
Department of Statistics, University of South Carolina[1]
Department of Transdisciplinary Studies, Seoul National University[2]
Tomocube Inc.[3]
Artificial Intelligence Graduate School, UNIST[4]
sungbin@unist.ac.kr

## Abstract

Bootstrapping has been a primary tool for ensemble and uncertainty quantification in machine learning and statistics. However, due to its nature of multiple training and resampling, bootstrapping deep neural networks is computationally burdensome; hence it has difficulties in practical application to the uncertainty estimation and related tasks. To overcome this computational bottleneck, we propose a novel approach called *Neural Bootstrapper* (NeuBoots), which learns to generate bootstrapped neural networks through single model training. NeuBoots injects the bootstrap weights into the high-level feature layers of the backbone network and outputs the bootstrapped predictions of the target, without additional parameters and the repetitive computations from scratch. We apply NeuBoots to various machine learning tasks related to uncertainty quantification, including prediction calibrations in image classification and semantic segmentation, active learning, and detection of out-of-distribution samples. Our empirical results show that NeuBoots outperforms other bagging based methods under a much lower computational cost without losing the validity of bootstrapping.

## 1 Introduction

Bootstrapping [7] or bagging [3] procedures have been commonly used as a primary tool in quantifying uncertainty lying on statistical inference, e.g. evaluations of standard errors, confidence intervals, and hypothetical null distribution. Despite its success in statistics and machine learning field, the naive use of bootstrap procedures in deep neural network applications has been less practical due to its computational intensity. Bootstrap procedures require evaluating a number of models; however, training multiple deep neural networks are infeasible in practice in terms of computational cost.

To utilize bootstrap for deep neural networks, we propose a novel bootstrapping procedure called **Neu**ral **Boots**trapper (NeuBoots). The proposed method is mainly motivated by Generative Bootstrap Sampler (GBS) [38], which trains a bootstrap generator by model parameterization based on Random Weight Bootstrapping (RWB, [37]) framework. For many statistical models, the idea of GBS is more theoretically valid than amortized bootstrap [31], which trains an implicit model to approximate the bootstrap distribution over model parameters. However, GBS is hardly scalable to modern deep neural networks containing millions of parameters.

Contrary to the previous method, the proposed method is effortlessly scalable and universally applicable to the various architectures. The key idea of NeuBoots is simple; multiplying bootstrap weights to the final layer of the backbone network and instead of model parameterization. Hence it outputs the bootstrapped predictions of the target without additional parameters and the repetitive

---

*Equal Contribution
†Corresponding Author

| | Standard Bootstrap [7] | MCDrop [13] | DeepEnsemble [24] | NeuBoots |
|---|:---:|:---:|:---:|:---:|
| Memory Efficiency | ✗ | ✓ | ✗ | ✓ |
| Fast Training | ✗ | ✓ | ✗ | ✓ |
| Fast Prediction | ✗ | ✗ | ✗ | ✓ |

**Table 1.1.** Computational comparison between bagging based uncertainty estimation methods in the view of memory efficiency and computational speed during the training and prediction step.

computations from scratch. NeuBoots outperforms the previous sampling-based methods [13, 24, 31] on the various uncertainty quantification related tasks with deep convolutional networks [17, 20, 22]. Throughout this paper, we show that NeuBoots has multiple advantages over the existing uncertainty quantification procedures in terms of memory efficiency and computational speed (see Table 1.1).

To verify the empirical power of the proposed method, we apply NeuBoots to a wide range of experiments related to uncertainty quantification and bagging. We apply the NeuBoots to prediction calibration, active learning, out-of-distribution (OOD) detection, semantic segmentation, and learning on imbalanced datasets. Notably, we test the proposed method on biomedical data of high-resolution, NIH3T3 data [5]. In Section 4, our results show that NeuBoots achieves at least comparable or better performance than the state-of-the-art methods in the considered applications.

## 2 Preliminaries

As preliminaries, we briefly review the standard bootstrapping [7] and introduce an idea of Generative Bootstrap Sampler (GBS, [38]), which is the primary motivation of the proposed method. Let $[m] := \{1, \ldots, m\}$ and denote a given training data by $\mathcal{D} = \{(X_i, y_i) : i \in [n]\}$, where each feature $X_i \in \mathcal{X} \subset \mathbb{R}^p$ and its response $y_i \in \mathbb{R}^d$. We denote the class of models $f : \mathbb{R}^p \to \mathbb{R}^d$ by $\mathcal{M}$. For the standard bootstrapping, we sample $B$ sets of bootstrap data $\mathcal{D}^{(b)} = \{(X_i^{(b)}, y_i^{(b)}) : i \in [n]\}$ with replacement for $b \in [B]$. For each bootstrap data $\mathcal{D}^{(b)}$, we define a loss functional $L$ on $f \in \mathcal{M}$:

$$L(f, \mathcal{D}^{(b)}) := \frac{1}{n} \sum_{i=1}^n \ell(f(X_i^{(b)}), y_i^{(b)}) \tag{2.1}$$

where $\ell : \mathbb{R}^d \times \mathbb{R}^d \to \mathbb{R}$ is an arbitrary loss function. Then we minimize (2.1) with respect to $f \in \mathcal{M}$ to obtain bootstrapped models: for $b \in [B]$,

$$\widehat{f}^{(b)} = \arg \min_{f \in \mathcal{M}} L(f, \mathcal{D}^{(b)}). \tag{2.2}$$

**Random Weight Bootstrapping** It is well-known that the standard bootstrap uses only (approximately) 63% of observations for each bootstrap evaluation [24]. To resolve this problem, we use Random Weight Bootstrapping (RWB, [37]), which reformulates (2.2) as a sampling of bootstrapping weights for a weighted loss functional. Let $\mathcal{W} = \{\mathbf{w} \in \mathbb{R}_+^n : \sum_{i=1}^n w_i = n\}$ be a dilated standard $(n-1)$-simplex. For $\mathbf{w} = (w_1, \ldots, w_n) \in \mathcal{W}$ and the original training data $\mathcal{D}$, we define the Weighted Bootstrapping Loss (WBL) functional on $f \in \mathcal{M}$ as follows:

$$L(f, \mathbf{w}, \mathcal{D}) := \frac{1}{n} \sum_{i=1}^n w_i \ell(f(X_i), y_i). \tag{2.3}$$

Then for any resampled dataset $\mathcal{D}^{(b)}$, there exists a unique $\mathbf{w} \in \mathcal{W}$ such that (2.1) matches to (2.3). This reformulation provides a relaxation method to consider full data set without any omission in bootstrapping. Precisely, as a continuous relaxation of the standard bootstrap, we use Dirichlet distribution [32]; $\mathbb{P}_{\mathcal{W}} = n \times \text{Dirichlet}(1, \ldots, 1)$, where $\mathbb{P}_{\mathcal{W}}$ is a probability distribution on the simplex $\mathcal{W}$. Hence RWB fully utilizes the observed data points, since sampled bootstrap weights $\mathbf{w} \sim \mathbb{P}_{\mathcal{W}}$ are strictly positive. Also, [34] showed that RWB achieves the same theoretical properties with these of the standard bootstrap i.e. $\mathbb{P}_{\mathcal{W}} = \text{Multinomial}(n; 1/n, \ldots, 1/n)$ in (2.3).

**Bootstrap Distribution Generator** Although RWB resolves the data discard problem, training multiple networks $\widehat{f}^{(1)}, \ldots, \widehat{f}^{(B)}$ remains a computational problem, and one has to store the parameters of every network for prediction. To reduce the computational bottlenecks, GBS [38] proposes a

procedure to train a generator function of bootstrapped estimators for parametric statistical models. The main idea of GBS is to parameterize the model parameter with bootstrap weight $\mathbf{w} \in \mathcal{W}$. When the GBS is applied to bootstrapping neural networks, it considers a *bootstrap generator* $g : \mathbb{R}^p \times \mathcal{W} \to \mathbb{R}^d$ with parameter $\theta(\mathbf{w})$, where $d$ is the total number of neural net parameters in $g$, so that $g(X, \mathbf{w}) = g_{\theta(\mathbf{w})}(X)$. Based on (2.3), we define a new WBL functional:

$$\mathcal{L}(g, \mathcal{D}) = \mathbb{E}_{\mathbf{w} \sim \mathbb{P}_{\mathcal{W}}}[L(g, \mathbf{w}, \mathcal{D})], \quad L(g, \mathbf{w}, \mathcal{D}) = \frac{1}{n} \sum_{i=1}^{n} w_i \ell(g(X_i, \mathbf{w}), y_i) \qquad (2.4)$$

Note that we use the Dirichlet distribution for $\mathbb{P}_{\mathcal{W}}$; hence the functional $\mathcal{L}(g, \mathcal{D})$ includes RWB procedure itself. Analogous to (2.2), we obtain the bootstrap generator $\widehat{g}$ by optimizing $\mathcal{L}(g, \mathcal{D})$:

$$\widehat{g} = \arg\min_{g \in \mathcal{M}} \mathcal{L}(g, \mathcal{D}) \qquad (2.5)$$

Then learned $\widehat{g}$ can generate bootstrap samples for given target data $X_*$ by plugging an arbitrary $\mathbf{w} \in \mathcal{W}$ into $\widehat{g}(X_*, \cdot)$. We refer to [38, Section 2] for detailed theoretical results on GBS.

**Block Bootstrapping**  The above bootstrap generator $g$ receives a bootstrap weight vector $\mathbf{w}$ of dimension $n$; hence its optimization via (2.5) would be hurdled when the number of data $n$ is large. Hence we utilize a block bootstrapping procedure to reduce the dimension of bootstrap weight vector. We allocate the index set $[n]$ to $S$ number of blocks. Let $u : [n] \to [S]$ denotes the assignment function. Then we impose the same value of weight on all elements in a block such as, $w_i = \alpha_s$ for $u(i) = s \in [S]$, where $\boldsymbol{\alpha} = (\alpha_1, \dots, \alpha_S) \sim S \times \text{Dirichlet}(1, \dots, 1)$. Instead of $\mathbf{w}$, we plug $\boldsymbol{\alpha}$ in $g(X, \cdot) = g_{\theta(\cdot)}(X)$ to generate bootstrap samples and compute the weighted loss function in (2.4):

$$\mathcal{L}(g, \mathcal{D}) = \mathbb{E}_{\boldsymbol{\alpha} \sim S \times \text{Dirichlet}(1, \dots, 1)} \left[ \frac{1}{n} \sum_{i=1}^{n} \alpha_{u(i)} \ell(g(X_i, \boldsymbol{\alpha}), y_i) \right] \qquad (2.6)$$

The above procedure asymptotically converges to the same target distribution where the conventional non-block bootstrap converges. See appendix A for more detailed procedure and proofs.

# 3   Neural Bootstrapper

Now we propose **Neu**ral **Boots**trapper (NeuBoots), which reduces computational complexity and memory requirement of the networks in the learning of bootstrapped distribution to being suitable for deep neural networks.

**How to implement the bootstrap generator $g$ for deep neural networks?**   One may consider directly applying GBS to existing deep neural networks by modeling a neural net $\theta(\cdot)$ that outputs the neural net parameters of $g$. However, this approach is computationally challenging due to the high-dimensionality of the output dimension of $\theta(\cdot)$ Indeed, [38] proposes an architecture which concatenates bootstrap weight vector to every layer of a given neural network (Figure 3.1(b)) and trains it with (2.6). However, the bagging performance of GBS gradually degrades as we applied it to the deeper neural networks. This may be because the information of bootstrap weights in the earlier layers less propagate since the target model reduces the parameters of the weights during the training.

## 3.1   Adaptive Block Bootstrapping

We found that the bootstrap weight in the final layer mainly affects the bootstrap performance of GBS. This fact motivates us to utilize the following adaptive block bootstrapping, which is the key idea of NeuBoots. Take a neural network $f_\theta \in \mathcal{M}$ with parameter $\theta$. Let $M_{\theta_1}$ and $F_{\theta_2}$ be the single-layer neural network in the final layer and the feature extractor of $f$, respectively, with parameter $\theta = (\theta_1, \theta_2)$, so we can decompose $f_\theta$ into $M_{\theta_1} \circ F_{\theta_2}$. Set $S := \dim(F_{\theta_2}(X))$ for the number of blocks for block bootstrapping. Then, we redefine bootstrap generator as follows:

$$g_\theta(X, \boldsymbol{\alpha}) := g_{(\theta, \boldsymbol{\alpha})}(X) = M_{\theta_1}(F_{\theta_2}(X) \odot \boldsymbol{\alpha}) \qquad (3.1)$$

where $\odot$ denotes an elementwise multiplication. Bootstrap generator (3.1) can also be trained with (2.6); hence optimized $\widehat{g}_\theta(X, \cdot)$ can generate the bootstrapped prediction as we plug $\boldsymbol{\alpha}$. This

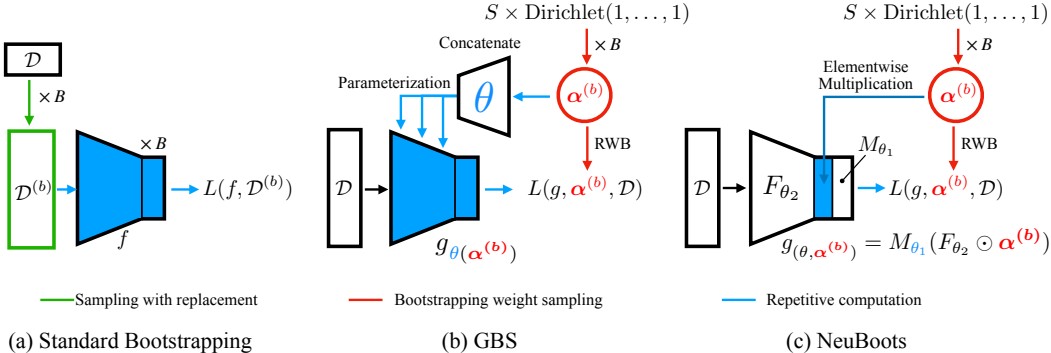

(a) Standard Bootstrapping      (b) GBS      (c) NeuBoots

**Figure. 3.1.** A comparison between the bootstrapping procedure of (a) standard bootstrapping [7], (b) GBS [38], and (c) NeuBoots. This figure is best viewed in color.

modification brings a computational benefit, since we can generate bootstrap samples quickly and memory-efficiently by reusing a priori computed tensor $F_{\theta_2}(X)$ without repetitive computation from scratch. See Figure 3.1 for the comparison between the previous methods and NeuBoots. In our empirical experience, the bootstrap evaluations over different groupings were consistent for all examined examples in this article.

**Training and Prediction**    At every epoch, we update the $\mathbf{w}_{\alpha} = \{\alpha_{u(1)}, \ldots, \alpha_{u(n)}\}$ randomly, and the expectation in (2.6) can be approximated by the average over the sampled weights. Considering the stochastic gradient descent (SGD) algorithms to update the parameter $\theta$ via mini-batch sequence $\{\mathcal{D}_k : \mathcal{D}_k \subset \mathcal{D}\}_{k=1}^{K}$, we plug mini-batch size of bootstrap weight vector $\{\alpha_{u(i)} : X_i \in \mathcal{D}_k\}$ in (2.6) without changing $\alpha$. Each element of $w_{\alpha}$ is not used repeatedly during the epoch, so the sampling and replacement procedures in Algorithm 1 are conducted once at the beginning of epoch. After we obtain the optimized network $\widehat{g}_{\theta}$, for the prediction, we use the generator $\widehat{g}_*(\cdot) = \widehat{g}_{\theta}(X_*, \cdot)$ for a given data point $X_*$. Then we can generate bootstrapped predictions by plugging $\alpha^{(1)}, \ldots, \alpha^{(B)}$ in the generator $\widehat{g}_*(\cdot)$, as described in Algorithm 2.

---

**Algorithm 1:** Training step in NeuBoots.

**Input**   : Dataset $\mathcal{D}$; epochs $T$; dimension of feature $S$; index function $u$; learning rate $\rho$.

1   Initialize neural network parameter $\phi^{(0)}$ and set $n := |\mathcal{D}|$.

2   **for** $t \in \{0, \ldots, T-1\}$ **do**

3      Sample $\alpha^{(t)} = \{\alpha_1^{(t)}, \ldots, \alpha_S^{(t)}\} \stackrel{\text{i.i.d.}}{\sim} S \times \text{Dirichlet}(1, \ldots, 1)$

4      Replace $\mathbf{w}_{\alpha}^{(t)} = \{\alpha_{u(1)}^{(t)}, \ldots, \alpha_{u(n)}^{(t)}\}$

5      Update $\theta^{(t+1)} \leftarrow \theta^{(t)} - \rho \nabla_{\theta} L(g_{(\theta, \alpha)}, \mathbf{w}_{\alpha}^{(t)}, \mathcal{D})\big|_{\theta = \theta^{(t)}}$.

---

**Algorithm 2:** Prediction step in NeuBoots.

**Input**   : Data point $X_* \in \mathbb{R}^p$; number of bootstrap sampling $B$.

1   Compute the feed-forward network $\widehat{g}_*(\cdot) = \widehat{g}_{\theta}(X_*, \cdot)$ a priori.

2   **for** $b \in \{1, \ldots, B\}$ **do**

3      Generate $\alpha^{(b)} \stackrel{\text{i.i.d.}}{\sim} S \times \text{Dirichlet}(1, \ldots, 1)$ and evaluate $\widehat{y}_*^{(b)} = \widehat{g}_*(\alpha^{(b)})$.

---

## 3.2   Discussion

**NeuBoots vs Standard Bootstrap**    To examine the approximation power of NeuBoots, we have measured the frequentist's coverage rate of the confidence bands (Figure 3.2.(a)). We estimate 95% confidence band for nonparametric regression function by using the NeuBoots, and compare

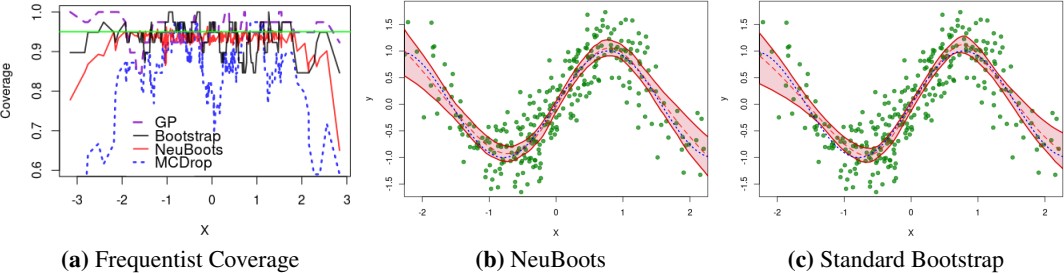

|  (a) Frequentist Coverage  |  (b) NeuBoots  |  (c) Standard Bootstrap  |

**Figure. 3.2.** (a) Frequentist coverage rate of the 95% confidence bands; (b) Curve fitting with different nonlinear funtions. 95% confidence band of the regression mean from NeuBoots; (c) the standard bootstrap. Each red dashed line indicates the mean, and the blue dotted lines show the true regression function.

it with credible bands (or confidence bands) evaluated by the standard bootstrap, Gaussian Process (GP) regression, and MCDrop [13]. We adopt Algorithm 1 to train the NeuBoots generator with 3 hidden-layers with 500 hidden-nodes for each layer. For the standard bootstrap, we train 1000 neural networks. The result shows the confidence band via NeuBoots stably covers the true regression function on each predictor value with almost 95% of frequency, which is compatible with the standard bootstrapping. In contrast, the coverage of the MCDrop is unstable and sometimes below 70%. This result indicates that the NeuBoots performs comparably with the standard bootstrapping in uncertainty quantification tasks.

**NeuBoots vs Amortized Bootstrapping** We applied NeuBoots to classification and regression experiments presented by the amortized bootstrap [31]. Indeed, every experiment demonstrates that NeuBoots outperforms the amortized bootstrap in bagging performance for various tasks: the rotated MNIST classification (Table 3.1), classification with different data points $N$ (Figure B.1), and regression on two datasets (Figure B.2). We remark the expected calibration error (ECE, [30]) score on the rotated MNIST is improved via NeuBoots from 15.00 to 2.98 by increasing the number of bootstrap sampling $B$.

| Methods | Test Error | | |
|---|---|---|---|
|  | $B = 1$ | $B = 5$ | $B = 25$ |
| Traditional Bootstrap | 22.57 | 19.68 | 18.57 |
| Amortized Bootstrap | **17.03** | 16.82 | 16.18 |
| NeuBoots | 17.94±0.74 | **14.98**±0.31 | **14.45**±0.31 |

**Table 3.1.** Rotated MNIST classification with different bootstrap sampling number $B$.

**NeuBoots vs Dropout** At first glance, NeuBoots is similar to Dropout in that the final neurons are multiplied by random variables. However, random weights imposed by the Dropout are lack of connection to the loss function nor the working model, while the bootstrap weights of the NeuBoots appears in the loss function (2.6) have explicit connections to the bootstrapping. We briefly verify the effect of the loss function on the 3-layers MLP with the different number of hidden variables 50, 100, and 200 for the image classification task on MNIST datasets. With batch normalization [21], we have applied Dropout with probability $p = 0.1$ only to the final layer of MLP. We measure the ECE, the negative log-likelihood (NLL), and the Brier score for comparisons. NeuBoots and Dropout records same accuracy. However, Figure B.3 shows that the NeuBoots is more feasible for confidence-aware learning and clearly outperforms the Dropout in terms of ECE, NLL, and the Brier score.

**Computation time and cost** As we mentioned earlier, the algorithm evaluates the network from scratch for only once to store the tensor $F_{\theta_2}(X_*)$, while the standard bootstrapping and MCDrop [13] need repetitive feed-forward propagations. To check this empirically, we measure the prediction time by ResNet-34 between NeuBoots and MCDrop on the test set of CIFAR-10 with Nvidia V100 GPUs. NeuBoots predicts $B = 100$ bootstrapping in 1.9s whereas MCDrop takes 112s to generate 100 outputs. Also, NeuBoots is computationally more efficient than the standard bootstrap and the sparse GP [40] (Figure 3.3).

| Method | Training Time | Test Time | Memory Usage |
|---|---|---|---|
| DeepEnsemble | $O(LK)$ | $O(LK)$ | $O(MK + I)$ |
| BatchEnsemble | $O(LK)$ | $O(LK)$ | $O(M + IK)$ |
| MIMO | $O(L + 2K)$ | $O(L + 2K)$ | $O(M + IK)$ |
| NeuBoots | $O(L)$ | $O(L + K)$ | $O(M + I)$ |

**Table 3.2.** A comparison of computational costs. We use the following notations: $L$ the number of layers, $K$ the number of bootstrapping (or ensemble), $M$ the parameter size of a single model, $I$ memory size of input data.

We also compare NeuBoots to MIMO [16] and BatchEnsemble [42] in terms of training, test, and memory complexities (see Table 3.2). Since NeuBoots does not require repetitive forward computations, its training and test costs are $O(L)$ and $O(L + K)$, respectively, less than $O(L+2K)$ of MIMO and $O(LK)$ of BatchEnsemble. Note that MIMO needs to copy input images as many as $K$ to supply into input layers. Even though it can compute in a single forward pass, it requires more memories to upload multiple inputs if the input data is high-dimensional (e.g., MRI/CT). The memory complexity of BatchEnsemble is similar to the one of MIMO since the memory usage of fast weights in BatchEnsemble is proportional to the dimension of input and output. This computational bottleneck is nothing to sneeze at in the application fields requiring on-device training

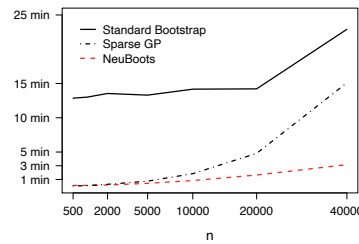

**Figure. 3.3.** Comparison of computational time with different numbers of training data $n$ for the example in Figure 3.2

or inference; however, the proposed method is free from such a problem since multiple computations occur only at the final layer. For quantitative comparisons, we refer to Appendix B.2.

**Diversity of predictions** Diversity of predictions has been a reliable measure to examine overfits and performance of uncertainty quantification for ensemble procedures [10, 35, 42]. In the presence of overfitting, it is expected that the diversity of different ensemble predictions would be minimal because the resulting ensemble models would produce similar predictions that are over-fitted towards the training data points. To examine the diversity performance of NeuBoots, we consider various diversity measures including ratio-error, Q-statistics, correlation coefficient, and prediction disagreement (see [1, 10, 35]). For the CIFAR-100 along with DenseNet-100, Table 3.3 summarizes the results. NeuBoots outperforms MCDrop in every metrics of diversity. Furthermore, NeuBoots shows comparable results with DeepEnsemble.

| Method | Ratio-error ($\uparrow$) | Q-stat ($\downarrow$) | Correlation ($\downarrow$) | Disagreement ($\uparrow$) |
|---|---|---|---|---|
| DeepEnsemble | **98.00** | **61.31** | 78.56 | 23.41 |
| MCDrop | 27.38 | 96.33 | 92.00 | 10.40 |
| NeuBoots | 93.79 | 63.95 | **76.11** | **32.20** |

**Table 3.3.** A comparison of diversity performances.

# 4 Empirical Studies

In this section, we conduct the wide range of empirical studies of NeuBoots for uncertainty quantification and bagging performance. We apply NeuBoots to prediction calibration, active learning, out-of-distribution detection, bagging performance for semantic segmentation, and learning on imbalanced dataset with various deep convolutional neural networks. Our code is open to the public[3].

## 4.1 Prediction Calibration

**Setting** We evaluate the proposed method on the prediction calibration for image classification tasks. We apply NeuBoots to image classification tasks on CIFAR and SVHN with ResNet-110 and DenseNet-100. We take $k = 5$ predictions of MCDrop and DeepEnsemble for calibration.

---

[3]https://github.com/sungbinlim/NeuBoots

For fair comparisons, we set the number of bootstrap sampling $B = 5$ as well, and fix the other hyperparameters same with baseline models. All models are trained using SGD with a momentum of 0.9, an initial learning rate of 0.1, and a weight decay of 0.0005 with the mini-batch size of 128. We use CosineAnnealing for the learning rate scheduler. We implement MCDrop and evaluates its performance with dropout rate $p = 0.2$, which is a close setting to the original paper. For Deep Ensemble, we utilize adversarial training and the Brier loss function [24] and cross-entropy loss function [2]. For the metric, we evaluate the error rate, ECE, NLL, and Brier score. We also compute each method's training and prediction times to compare the relative speed based on the baseline.

**Results** See Table B.3 and B.4 for empirical results. NeuBoots generally show a comparable calibration ability compared to MCDrop and DeepEnsemble. Figure 4.1.(a) shows the reliability diagrams of ResNet-110 and DenseNet-100 on CIFAR-100. We observe that NeuBoots secures accuracy and prediction calibration in the image classification tasks with ResNet-110 and DenseNet-100. NeuBoots is faster in prediction than MCDrop and DeepEnsemble at least three times. Furthermore, NeuBoots shows faster in training than Deep Ensemble at least nine times. This gap increases as the number of predictions $k$ increases. It concludes that NeuBoots outperforms MCDrop and is comparable with DeepEnsemble in calibrating the prediction with the relatively faster prediction.

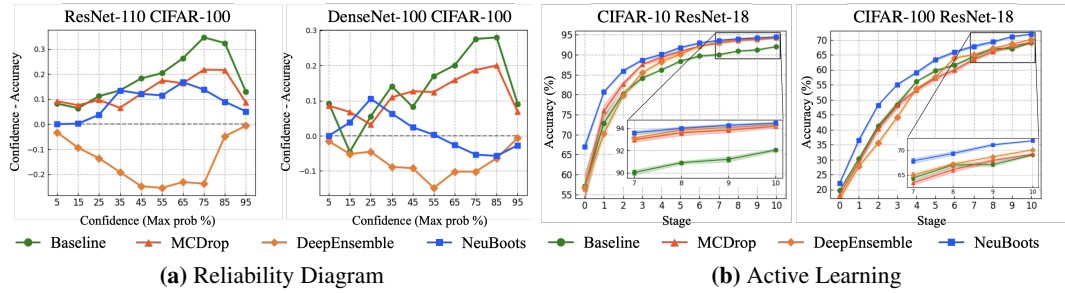

**(a)** Reliability Diagram    **(b)** Active Learning

**Figure. 4.1.** (a) Comparison of reliability diagrams for ResNet-110 and DenseNet-100 on CIFAR-100. Confidence is the value of the maximal softmax output. A dashed black line represents a perfectly calibrated prediction. Points below this line indicate to under-confident predictions, whereas points above the line mean overconfident predictions. (b) Actice learning performance on CIFAR-10 (left) and CIFAR-100 (right) with Random, MCDrop, and NeuBoots. Curves are averaged over five runs and shaded regions denote the confidence intervals.

## 4.2 Active Learning

**Setting** We evaluate the NeuBoots on the active learning with ResNet-18 architecture on CIFAR. For a comparison, we consider MCDrop and DeepEnsemble with entropy-based sampling and random sampling. We follow an ordinary process to evaluate the performance of active learning (see [29]). Initially, a randomly sampled 2,000 labeled images are given, and we train a model. Based on the uncertainty estimation of each model, we sample 2,000 additional images from the unlabeled dataset and add to the labeled dataset for the next stage. We continue this process ten times for a single trial and repeat five trials for each model.

**Results** Figure 4.1.(b) shows the sequential performance improvement on CIFAR-10 and CIFAR-100. Note that CIFAR-100 is more challenging dataset than CIFAR-10. Both plots demonstrate that NeuBoots is superior to the other sampling methods in the active learning task. NeuBoots records 71.6% accuracy in CIFAR-100 and 2.5% gap with MCDrop and DeepEnsemble. Through the experiment, we verify that NeuBoots has a significant advantage in active learning.

## 4.3 Out-of-Distribution Detection

**Setting** As an important application of uncertainty quantification, we have applied NeuBoots to detection of out-of-distribution (OOD) samples. The setting for OOD is based on the Mahalanobis method [26]. At first, we train ResNet-34 for the classification task only using the training set of the CIFAR-10 (in-distribution). Then, we evaluate the performance of NeuBoots for OOD detection both in the test sets of in-distribution dataset and the SVHN (out-of-distribution). Using a separate

validation set from the testsets, we train a logistic regression based detector to discriminate OOD samples from in-distribution dataset. For the input vectors of the OOD detector, we extract the following four statistics regarding logit vectors: the max of predictive mean vectors, the standard deviation of logit vectors, expected entropy, and predictive entropy, which can be computed by the sampled output vectors of NeuBoots. To evaluate the performance of the detector, we measure the true negative rate (TNR) at 95% true positive rate (TPR), the are under the receiver operating characteristic curve (AUROC), the area under the precision-recall curve (AUPR), and the detection accuracy. For comparison, we examine the baseline method [19], MCDrop, DeepEnsemble [24], DeepEnsemble_CE (trained with cross-entropy loss) [2], ODIN [27], and Mahalanobis [26].

| Method | TNR at TPR 95% | AUROC | Detection Accuracy | AUPR In | AUPR Out |
|---|---|---|---|---|---|
| Baseline | 32.47 | 89.88 | 85.06 | 85.4 | 93.96 |
| MCDrop | 51.4 | 92.01 | 89.46 | 86.82 | 95.41 |
| DeepEnsemble [24] | 56.7 | 91.85 | 88.91 | 81.66 | 95.46 |
| DeepEnsemble_CE [2] | 48.5 | 92.29 | 90.48 | 86.33 | 95.49 |
| ODIN | 86.55 | 96.65 | 91.08 | 92.54 | 98.52 |
| Mahalanobis | 54.51 | 93.92 | 89.13 | 91.54 | 98.52 |
| Mahalanobis + Calibration | 96.42 | **99.14** | 95.75 | **98.26** | 99.6 |
| NeuBoots | 89.40 | 97.26 | 93.80 | 93.97 | 98.86 |
| NeuBoots + Calibration | **99.00** | **99.14** | **96.52** | 97.78 | **99.68** |

**Table 4.1.** OOD detection. All values are percantages and the best results are indicated in bold.

**Results** Table 4.1 shows NeuBoots significantly outperform the baseline method [19], DeepEnsemble [2, 24], and ODIN [27] without any calibration technique in OOD detection. Furthermore, with the input pre-processing technique studied in [27], NeuBoots is superior to Mahalanobis [26] in most metrics, which employs both the feature ensemble and the input pre-processing for the calibration techniques. This validates NeuBoots can discriminate OOD samples effectively. In order to see the performance change of the OOD detector concerning the bootstrap sample size, we evaluate the predictive standard deviation estimated by the proposed method for different $B \in \{2, 5, 10, 20, 30\}$. Figure B.5 illustrates that the NeuBoots successfully detects the in-distribution samples (top row) and the out-of-distribution samples (bottom row).

### 4.4 Bagging Performance for Semantic Segmentation

**Setting** To demonstrate the applicability of NeuBoots to different computer vision tasks, we validate NeuBoots on PASCAL VOC 2012 semantic segmentation benchmark [9] with DeepLab-v3 [4] based on the backbone architectures of ResNet-50 and ResNet-101. We modify the final $1 \times 1$ convolution layer after the Atrous Spatial Pyramid Pooling (ASPP) module by multiplying the channel-wise bootstrap weights. This is a natural modification of the segmentation architecture analogous to the fully connected layer of the networks for classification tasks. Additionally, we apply NeuBoots to real 3D image segmentation task on commercial ODT microscopy NIH3T3 [5] dataset, which is challenging for not only models but also human due to the $512 \times 512 \times 64$ sized large resolution and endogenous cellular variability. We use two different U-Net-like models for this 3D image segmentation task, which are U-ResNet and SCNAS. We simply amend the bottleneck layer in the same way as the 2D version. Same as an image classification task, we set $B = 5$ and $k = 5$. For the remaining, we follow the usual setting.

**Results** Table 4.2 shows NeuBoots significantly improves mean IoU and ECE compared to the baseline. Furthermore, similar to the image classification task, NeuBoots records faster prediction time than MCDrop and DeepEnsemble. This experiment indeed verifies that NeuBoots can be applied to the wider scope of computer vision tasks beyond image classification.

### 4.5 Imbalanced Dataset

**Setting** To validate the efficacy for the imbalanced dataset, we have applied NeuBoots to two imbalance sets, the imbalanced CIFAR-10 and the white blood cell dataset with ResNet-18. To

| Dataset | Architecture | Method | mIoU(%) | ECE(%) | Relative Prediction Time |
|---------|--------------|--------|---------|--------|--------------------------|
| **2D**
(PASCAL VOC [9]) | ResNet-50 | Baseline | 84.57±0.72 | 15.35±0.21 | 1.0 |
| | | MCDrop | 87.81±1.83 | 6.6±0.1 | 5.4 |
| | | DeepEnsemble [24] | 90.09±0.61 | 17.31±0.74 | 5.5 |
| | | DeepEnsemble_CE [2] | 86.95±0.57 | 12.36±0.53 | 5.5 |
| | | NeuBoots | **90.14**±2.17 | **6.00**±0.1 | **2.7** |
| | ResNet-101 | Baseline | 85.35±0.23 | 15.49±0.44 | 1.0 |
| | | MCDrop | 88.08±1.80 | 6.48±0.08 | 5.3 |
| | | DeepEnsemble [24] | 90.40±0.11 | 17.94±0.03 | 5.3 |
| | | DeepEnsemble_CE [2] | 87.48±0.09 | 11.52±0.02 | 5.3 |
| | | NeuBoots | **90.56**±1.71 | **6.14**±0.11 | **2.5** |
| **3D**
(NIH3T3 [5]) | U-ResNet | Baseline | 61.54±1.14 | 1.85±0.19 | 1.0 |
| | | MCDrop | 64.15±0.48 | 1.53±0.09 | 5.5 |
| | | DeepEnsemble [24] | 59.71±1.82 | 1.78±0.29 | 5.5 |
| | | DeepEnsemble_CE [2] | 65.71±1.69 | **0.94**±0.24 | 5.5 |
| | | NeuBoots | **67.78**±1.01 | 1.67±0.19 | **3.5** |
| | SCNAS [22] | Baseline | 67.52±1.95 | 1.45±0.19 | 1.0 |
| | | MCDrop | 65.37±1.13 | 0.64±0.17 | 5.2 |
| | | DeepEnsemble [24] | 60.04±2.11 | 1.39±0.05 | 5.3 |
| | | DeepEnsemble_CE [2] | 68.66±2.58 | 0.83±0.09 | 5.3 |
| | | NeuBoots | **70.80**±1.58 | **0.63**±0.16 | **2.1** |

**Table 4.2.** Semantic segmentation. The best results are indicated in bold.

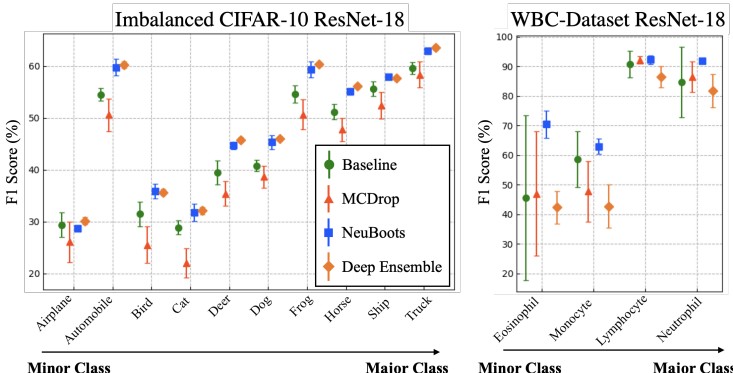

**Figure. 4.2.** Comparisons of classification power for imbalance datasets. The minor class refers to the class with the least number of samples, and the major class refers to the highest number of samples.

conduct an imbalanced CIFAR-10, we randomly sampled from the training dataset of CIFAR-10 to follow a different distribution for each class, and the distribution is [*50, 100, 150, 200, 250, 300, 350, 400, 450, 500*] for [*airplane, automobile, bird, cat, deer, dog, frog, horse, ship, truck*]. The white blood cell dataset was acquired using a commercial ODT microscopy, and each image is a grayscale of $80 \times 80$. The dataset comprises four types of white blood cells, and the training distribution is [*144, 281, 2195, 3177*] for [*eosinophil, monocyte, lymphocyte, neutrophil*]. ResNet-18 model and MCDrop extension are used as a baseline and comparative model with the same settings as Section 4.2, respectively. We measure the F1 Score for each class for evaluation.

**Results**    Comparing the performance of Baseline, MCDrop, and DeepEmsenble, NeuBoots performs better on both imbalanced CIFAR-10 and the white blood cell dataset, as shown in Figure 4.2. Especially, NeuBoots outperforms for eosinophil identification, the class with the lowest number of samples in the white blood cell dataset, with low variance. This result shows that the NeuBoots boosts the prediction power for the fewer sampled classes with high stability via simple implementation.

## 5 Related Work

**Bootstrapping Neural Network**    Since [7] first proposed the nonparametric bootstrapping to quantify uncertainty in general settings, there has been a rich amount of literature that investigate theoretical advantages of using bootstrap procedures for parametric models [8, 14, 15]. For nerural networks, [12] investigated bootstrap consistency of one-layered MLP under some strong regularity conditions. [36] considered using a conventional nonparametric bootstrapping to robustify classifiers under noisy labeling. However, due to the nature of repetitive computations, its practical application to large-sized data sets is not trivial. [31] proposed an approximation of bootstrapping for neural network by applying amortized variational Bayes. Despite its computational efficiency, the armortized bootstrap does not induce the exact target bootstrap distribution, and its theoretical justification is lacking. Recently, [25] proposes a bootstrapping method for neural processes. They utilized residual bootstrap to resolve the data discard problem, but their approach is not scalable since it requires multiple encoder computations.

**Ensemble Methods**    Various advances of neural net ensembles have been made to improve computational efficiency and uncertainty quantification performance. Havasi et al. [16] introduces Multiple Input Multiple Output (MIMO), that approximates independent neural nets by imposing multiple inputs and outputs, and Wen et al. [42] proposes a low-rank approximation of ensemble networks, called BatchEnsemble. Latent Posterior Bayes NN (LP-BNN, [11]) extends the BatchEnsemble to a Bayesian paradigm imposing a VAE structure on the individual low-rank factors, and the LP-BNN outperforms the MIMO and the BatchEnsemble in prediction calibration and OOD detection, but its computational burden is heavier than that of the BatchEnsemble. Stochastic Weight Averaging Gaussian (SWAG, [28]) computes the posterior of the base neural net via a low-rank approximation with a batch sampling. Even though these strategies reduces the computational cost to train each ensemble network, unlike NeuBoots, they still demand multiple optimizations, and its computational cost linearly increases as the ensemble size grows up.

**Uncertainty Estimation**    There are numerous approaches to quantify the uncertainty in predictions of NNs. Deep Confidence [6] proposes a framework to compute confidence intervals for individual predictions using snapshot ensembling and conformal prediction. Also, a calibration procedure to approximate a confidence interval is proposed based on Bayesain neural networks [23]. Gal and Ghahramani [13] proposes MCDrop which captures model uncertainty casting dropout training in neural networks as an approximation of variational Bayes. Smith and Gal [39] examines various measures of uncertainty for adversarial example detection. Lakshminarayanan et al. [24] proposes a non-Bayesian approach, called DeepEnsemble, to estimate predictive uncertainty based on ensembles and adversarial training. Compared to DeepEnsemble, NeuBoots does not require adversarial training nor learning multiple models.

## 6 Conclusion

We introduced a novel and scalable bootstrapping method, NeuBoots, for neural networks. We applied it to the wide range of machine learning tasks related to uncertainty quantification; prediction calibration, active learning, out-of-distribution detection, and imbalanced datasets. NeuBoots also demonstrates superior bagging performance over semantic segmentation. Our empirical studies show that NeuBoots attains significant potential in quantifying uncertainty for large-sized applications, such as biomedical data analysis with high-resolution. As a future research, one can apply NeuBoots to natural language processing tasks using Transformor [41].

## 7 Acknowledgement

The authors specially thanks to Dr. Hyokun Yun for his fruitful comments. Minsuk Shin would like to acknowledge support from the National Science Foundation (NSF-DMS award #2015528). This work was also supported by Institute of Information & communications Technology Planning & Evaluation(IITP) grant funded by the Korea government(MSIT) (No.2020-0-01336, Artificial Intelligence Graduate School Program(UNIST)) and National Research Foundation of Korea(NRF) funded by the Korea government(MSIT)(2021R1C1C1009256).

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
