# A   Block Bootstrapping

## Implementation

Let $I_1, \ldots, I_S$ denotes the index sets of exclusive $S$ blocks where $S \ll n$. We allocate the index of data $[n]$ to each block $I_1, \ldots, I_S$ by the stratified sampling to balance among classes. Let index function $u : [n] \to [S]$ denotes such assignment so $u(i) = s$ if $i \in I_s$. Then, given some weight distribution $H_{\boldsymbol{\alpha}}$ on $\mathcal{W}_S \subset \mathbb{R}_+^S$, we impose the same value of weight on all elements in a block such as, $w_i = \alpha_{u(i)}$ for $i \in [n]$, where $\boldsymbol{\alpha} = \{\alpha_1, \ldots, \alpha_S\} \sim H_{\boldsymbol{\alpha}}$. We write an allocated weight vector as $\mathbf{w}_{\boldsymbol{\alpha}} = \{\alpha_{u(1)}, \ldots, \alpha_{u(n)}\} \in \mathcal{W}_n$. Similar with GBS, setting $H_{\boldsymbol{\alpha}} = S \times \text{Dirichlet}(1, \ldots, 1)$ induces a block version of the RWB, and imposing $H_{\boldsymbol{\alpha}} = \text{Multinomial}(S; 1/S, \ldots, 1/S)$ results in a block nonparametric bootstrap. We remark that the Dirichlet distribution with a uniform parameter of one can be easily approximated by independent exponential distribution. That is, $z_i / \sum_{k=1}^n z_k \sim \text{Dirichlet}(1, \ldots, 1)$ for independent and identically distributed $z_i \sim \text{Exp}(1)$. Due to the fact that $\sum_{i=k}^n z_k / n \approx 1$ by the law of large number for a moderately large $n$, $n^{-1} \times \{z_1, \ldots, z_n\}$ approximately follows the Dirichlet distribution. This property is convenient in a sense that we do not need to consider the dependence structure in $\mathbf{w}$, and simply generate independent samples from $\text{Exp}(1)$ to sample the bootstrap weight. We use this block bootstrap as a default of the NeuBoots in sequel. The proposed procedure asymptotically converges towards the same target distribution where the conventional non-block bootstrap converges to, and under some mild regularity conditions. Theoretically, the block bootstrap asymptotically approximates the non-blocked bootstrap well as the number of blocks $S$ increases as $n \to \infty$ (see Theorem A.1).

## Asymptotics of Block Bootstrap

We shall rigorously investigate asymptotic equivalence between the blocked bootstrap and the non-blocked bootstrap. To ease the explanation for theory, we introduce some notation here. We distinguish a random variable $Y_i$ and its observed value $y_i$, and we assume that the feature $X_1, X_2, \ldots$ is deterministic. the Euclidean norm is denoted by $\|\cdot\|$, and the norm of a $L_2$ space is denoted by $\|\cdot\|_2$. Also, to emphasize that the bootstrap weight $\mathbf{w}$ depends on $n$, we use $\mathbf{w}_n$. Let $Y_1, Y_2, \ldots$ be i.i.d. random variables from the probability measure space $(\Omega, \mathcal{F}, \mathbb{P}_0)$. We denote the empirical probability measure by $\widehat{\mathbb{P}}_n := \sum_{i=1}^n \delta_{Y_i}/n$, where $\delta_x$ is a discrete point mass at $x \in \mathbb{R}$, and let $\mathbb{P}g = \int g d\mathbb{P}$, where $\mathbb{P}$ is a probability measure and $g$ is a $\mathbb{P}$-measurable function. Suppose that $\sqrt{n}(\widehat{\mathbb{P}}_n - \mathbb{P}_0)$ weakly converges to a probability measure $\mathbb{T}$ defined on some sample space and its sigma field $(\Omega', \mathcal{F}')$. In the regime of bootstrap, what we are interested in is to estimate $\mathbb{T}$ by using some weighted empirical distribution that is $\widehat{\mathbb{P}}_n^* = \sum_{i=1}^n w_i \delta_{Y_i}$, where $w_1, w_2, \ldots$ is an i.i.d. weight random variable from a probability measure $\mathbb{P}_{\mathbf{w}}$. In the same sense, the probability measure acts on the block bootstrap is denoted by $\mathbb{P}_{\mathbf{w}_{\boldsymbol{\alpha}}}$. We state a primary condition on bootstrap theory as follows:

$$\sqrt{n}(\widehat{\mathbb{P}}_n g - \mathbb{P}_0 g) \to \mathbb{T}g \quad \text{for } g \in \mathcal{D} \quad \text{and} \quad \mathbb{P}_0 g_{\mathcal{D}}^2 < \infty, \tag{A.1}$$

where $\mathcal{D}$ is a collection of some continuous functions of interest, and $g_{\mathcal{D}}(\omega) = \sup_{g \in \mathcal{D}} |g(\omega)|$ is the envelope function on $\mathcal{D}$. This condition means that there exists a target probability measure and the functions of interest should be square-bounded.

Based on this condition, the following theorem states that the block bootstrap asymptotically induces the same bootstrap distribution with that of non-block bootstrap. All proofs of theorems are deferred to the supplementary material.

**Theorem A.1.** *Suppose that* (A.1) *holds and* $\{\alpha_1, \ldots, \alpha_S\}^{\mathsf{T}} \sim S \times Dirchlet(1, \ldots, 1)$ *with* $w_i = \alpha_{u(i)}$. *We assume some regularity conditions introduced in the supplementary material, and also assume* $S \to \infty$ *as* $n \to \infty$. *Then, for a* $r_n$ *such that* $\|\widehat{f} - f_0\|_2 = O_{\mathbb{P}_{\mathbf{w}}}(\zeta_n r_n^{-1})$ *for any diverging sequence* $\zeta_n$,

$$\sup_{x \in \mathcal{X}, U \in \mathcal{B}} \left| \mathbb{P}_{\mathbf{w}} \left\{ r_n(\widehat{f}_{\mathbf{w}}(x) - \widehat{f}(x)) \in U \right\} - \mathbb{P}_{\mathbf{w}_{\boldsymbol{\alpha}}} \left\{ r_n(\widehat{f}_{\mathbf{w}_{\boldsymbol{\alpha}}}(x) - \widehat{f}(x)) \in U \right\} \right| \to 0, \tag{A.2}$$

*in* $\mathbb{P}_0$-*probability, where* $\mathcal{B}$ *is the Borel sigma algebra.*

Recall that the notation is introduced in Section 2. [34] showed that the following conditions on the weight distribution to derive bootstrap consistency for general settings:

**W1.** $\mathbf{w}_n$ is exchangeable for $n = 1, 2, \ldots$.
**W2.** $w_{n,i} \geq 0$ and $\sum_{i=1}^{n} w_{n,i} = n$ for all $n$.
**W3.** $\sup_n \|w_{n,1}\|_{2,1} < \infty$, where $\|w_{n,1}\|_{2,1} = \int \sqrt{\mathbb{P}_{\mathbf{w}}(w_{n,1} \geq t)} dt$.
**W4.** $\lim_{\lambda \to \infty} \limsup_{n \to \infty} \sup_{t \geq \lambda} t^2 \mathbb{P}_{\mathbf{w}}(w_{n,1} \geq t) = 0$.
**W5.** $n^{-1} \sum_{i=1}^{n} (w_{n,i} - 1)^2 \to 1$ in probability.

Under **W1**-**W5**, combined with (A.1), showed that $\sqrt{n}(\widehat{\mathbb{P}}_n^* - \widehat{\mathbb{P}}_n)$ weakly converges to $\mathbb{T}$. It was proven that the Dirichlet weight distribution satisfies **W1**-**W5**, and we first show that the Dirichlet weight distribution for the blocks also satisfies the condition. Then, the block bootstrap of the empirical process is also consistent when the classical bootstrap of the empirical process is consistent.

Since the block bootstrap randomly assigns subgroups, the distribution of $\mathbf{w}_n$ is exchangeable, so the condition **W1** is satisfied. The condition **W2** and **W3** are trivial. Since a Dirichlet distribution with a unit constant parameter can be approximated by a pair of independent exponential random variables; i.e $\{z_1/\sum_{i=1}^{S} z_i, \ldots, z_S/\sum_{i=1}^{S} z_i\} \sim Dir(1, \ldots, 1)$, where $z_i \overset{i.i.d.}{\sim} \exp(1)$. Therefore, $S \times Dir(1, \ldots, 1) \approx \{z_1, \ldots, z_S\}$, if $S$ is large enough. This fact shows that $t^2 \mathbb{P}_{\mathbf{w}}(w_{n,1} \geq t) \approx t^2 \mathbb{P}_z(z_1 \geq t)$, and it follows that $\mathbb{P}_z(z_1 \geq t) = \exp(-t)$, so **W4** is shown. The condition **W5** is trivial by the law of large number. Then, under **W1**-**W5**, Theorem 2.1 in [34] proves that

$$\sqrt{n}(\widehat{\mathbb{P}}_n^* - \widehat{\mathbb{P}}_n) \Rightarrow \mathbb{T}, \tag{A.3}$$

where the convergence "$\Rightarrow$" indicates weakly convergence.

We denote the true neural net parameter by $\phi_0$ such that $f_0 = f_{\phi_0}$, where $f_0$ is the true function that involves in the data generating process, and $\widehat{\phi}$ and $\widehat{\phi}_{\mathbf{w}}$ are the minimizers of the (**??**) for one-vector (i.e. $\mathbf{w} = (1, \ldots, 1)$) and given $\mathbf{w}$, respectively. This indicates that $\widehat{f} = f_{\widehat{\phi}}$ and $\widehat{f}_{\mathbf{w}} = f_{\widehat{\phi}_{\mathbf{w}}}$. Then, our objective function can be expressed as minimizing $\widehat{P}_n L(f_\phi(X), y)$ with respect to $\phi$. We further assume that

**A1.** the true function belongs to the class of neural network, i.e. $f_0 \in \mathcal{F}$.

**A2.** $\sup_{x \in \mathcal{X}, U \in \mathcal{B}} \left| \mathbb{P}_{\mathbf{w}} \left\{ r_n(\widehat{f}_{\mathbf{w}}(x) - \widehat{f}(x)) \in U \right\} - \mathbb{P}_0 \left\{ r_n(\widehat{f}(x) - f_0(x)) \in U \right\} \right| \to 0$,

in $\mathbb{P}_0$-probability, where $f_0$ is the true function that involves in the data generating process.

**A3.** Suppose that $\sum_{i=1}^{n} \frac{\partial}{\partial \phi} L(f_{\widehat{\phi}}(X_i), y_i) = 0$, $\sum_{i=1}^{n} \frac{\partial}{\partial \phi} w_i L(f_{\widehat{\phi}_{\mathbf{w}}}(X_i), y_i) = 0$ for any $\mathbf{w}$, and $\mathbb{E}_0[\frac{\partial}{\partial \phi} L(f_{\phi_0}(X), y)] = 0$.

**A4.** $\mathcal{H}$ is in $\mathbb{P}_0$-Donsker family, where $\mathcal{H} = \{\frac{\partial}{\partial \phi} L(f_\phi(\cdot), \cdot) : \phi \in \Phi\}$; i.e. $\sqrt{n}(\widehat{\mathbb{P}}_n g - \widehat{\mathbb{P}}_0 g) \to \mathbb{T}g$ for $g \in \mathcal{H}$ and $\mathbb{P}_0 g_{\mathcal{H}}^2 < \infty$.

These conditions assume that the classical weighted bootstrap is consistent, and a rigorous theoretical investigation of this consistency is non-existent at the current moment. However, we remark that the main purpose of this theorem is to show that the considered block bootstrap induces asymptotically the same result from the classical non-block bootstrap so that the use of the block bootstrap is at least asymptotically equivalent to the classical counterpart. In this sense, it is reasonable to assume that the classical bootstrap is consistent.

Then, it follows that

$$\sup_{x \in \mathcal{X}, U \in \mathcal{B}} \left| \mathbb{P}_{\mathbf{w}} \left\{ r_n(\widehat{f}_{\mathbf{w}}(x) - \widehat{f}(x)) \in U \right\} - \mathbb{P}_{\mathbf{w}_\alpha} \left\{ r_n(\widehat{f}_{\mathbf{w}_\alpha}(x) - \widehat{f}(x)) \in U \right\} \right|$$

$$\leq \sup_{x \in \mathcal{X}, U \in \mathcal{B}} \left| \mathbb{P}_{\mathbf{w}} \left\{ r_n(\widehat{f}_{\mathbf{w}}(x) - \widehat{f}(x)) \in U \right\} - \mathbb{P}_0 \left\{ r_n(\widehat{f}(x) - f_0(x)) \in U \right\} \right|$$

$$+ \sup_{x \in \mathcal{X}, U \in \mathcal{B}} \left| \mathbb{P}_{\mathbf{w}_\alpha} \left\{ r_n(\widehat{f}_{\mathbf{w}_\alpha}(x) - \widehat{f}(x)) \in U \right\} - \mathbb{P}_0 \left\{ r_n(\widehat{f}(x) - f_0(x)) \in U \right\} \right|.$$

The first part in the right-hand side of the inequality converges to 0 by **A1**. Also, the second part also converges to 0. That is because the empirical process of the block weighted bootstrap is asymptotically equivalent to the classical RWB, so **A2** and **A3** guarantees that the asymptotic behavior of the bootstrap solution should be consistent as the classical counterpart does. $\square$

# B Additional Experimental Results

## B.1 NeuBoots vs Amortized Bootstrapping

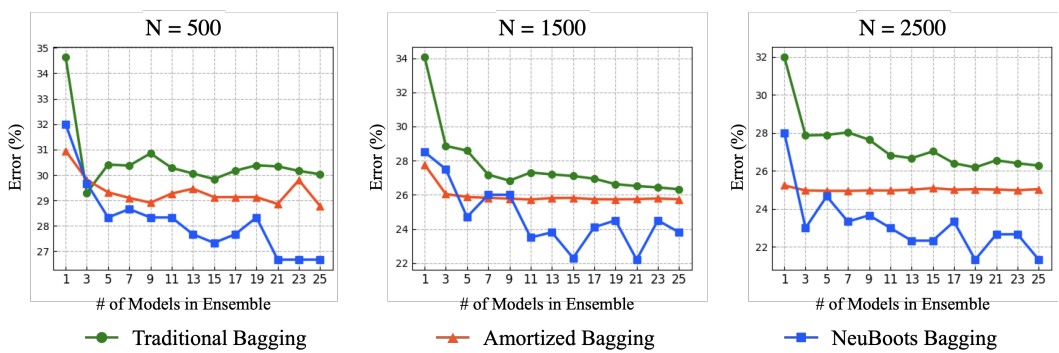

**Figure. B.1.** Comparison between standard bootstrapping, amortized bootstrapping [31], and NeuBoots in Classification.

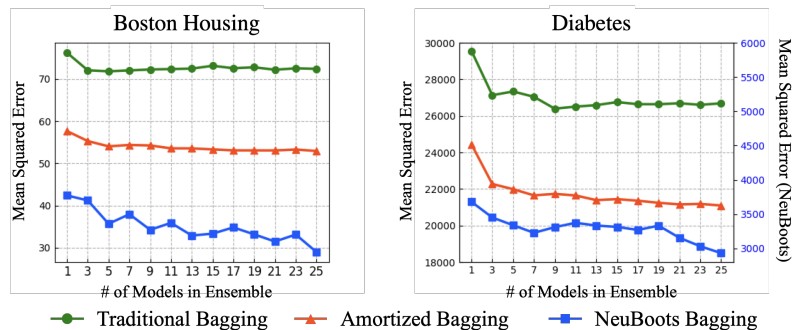

**Figure. B.2.** Comparison between standard bootstrapping, amortized bootstrapping[31], and NeuBoots in Regression.

## B.2 Computation time and costs

We supplement the comparison between MIMO, BatchEnsemble, and NeuBoots. We estimate training time and prediction during one epoch and measure the memory cost of those methods in ResNet-110 on CIFAR-100.

| Method | Training Time (sec) | Test Time (sec) | Memory Usage (mb) |
|---|---|---|---|
| Baseline | 29.1 | 1.69 | 1605 |
| MCDrop | 29.2 | 8.45 | 1605 |
| DeepEnsemble | 145.5 | 8.45 | 7600 |
| BatchEnsemble | 217.1 | 6.59 | 2474 |
| MIMO | 30.2 | 1.87 | 1605 |
| NeuBoots | 29.2 | 1.71 | 1605 |

**Table B.1.** A comparison of training time, test time, and memory cost.

We verified that NeuBoots benefits computational efficiency compared to BatchEnsemble. Interestingly, the above experiment shows that the computational cost of MIMO is similar to that of NeuBoots[4]. This result is because we used a convolutional neural network that benefits channel parallelism thanks to GPU. However, if we apply MIMO to MLP, the result is different: for 5-layer

---

[4]The memory comparison between NeuBoots and MIMO seems unclear due to the small resolution of CIFAR. We observed NeuBoots has lower memory cost than MIMO if we change the model from ResNet to DenseNet.

MLP ($K = 3$), NeuBoots takes 7.57 seconds for one epoch training, MIMO takes 41.09 seconds, as we expected. Also, we observed that growing the number of the ensemble makes the proportion gap increase: for $K = 5$, NeuBoots takes 7.68 seconds, MIMO takes 62.47 seconds. Furthermore, the below table shows that MIMO requires more memory cost in test time as the number of ensembles increases, as we expected in Table 3.2.

| Method | $K = 1$ | $K = 2$ | $K = 3$ | $K = 4$ | $K = 5$ |
|---|---|---|---|---|---|
| MIMO | 1515 | 1633 | 1741 | 1885 | 2067 |
| NeuBoots | 1515 | 1517 | 1517 | 1517 | 1517 |

**Table B.2.** A comparison between MIMO and NeuBoots in memory cost (mb) as the number of ensemble increases.

## B.3 Prediction Calibration

| Architecture | Method | Relative Training Time | Relative Prediction Time | Error Rate(%) | ECE(%) | NLL(%) | Brier Score(%) |
|---|---|---|---|---|---|---|---|
| ResNet-110 | Baseline | 1.0 | 1.0 | 26.69±0.35 | 16.43±0.15 | 14.19±0.71 | 42.09±0.51 |
| | MCDrop | 1.1 | 5.0 | 26.45 ±0.08 | 13.65±1.25 | 13.16±0.64 | 40.46±0.30 |
| | DeepEnsemble [24] | 9.5 | 5.0 | 34.84±0.21 | 27.33±4.92 | 18.69±1.44 | 56.12±3.02 |
| | DeepEnsemble [2] | 5.0 | 5.0 | **24.28**±0.11 | **4.74**±0.17 | **7.05**±0.28 | **28.29**±0.12 |
| | NeuBoots | **1.1** | **1.2** | 26.53±0.19 | 8.13±0.28 | 15.68±0.31 | 39.31±0.64 |
| DenseNet-100 | Baseline | 1.0 | 1.0 | 24.02±0.3 | 12.38±0.21 | 10.93±0.34 | 36.40±0.63 |
| | MCDrop | 1.1 | 5.0 | 23.88±0.09 | 9.49±0.35 | 10.22±0.86 | 34.94±0.67 |
| | DeepEnsemble [24] | 9.5 | 5.0 | 25.51±0.24 | 6.67±5.06 | 9.66±0.24 | 35.33±1.21 |
| | DeepEnsemble [2] | 5.0 | 5.0 | **20.16**±0.21 | 4.74±0.42 | **7.07**±0.14 | **30.29**±0.12 |
| | NeuBoots | **1.1** | **1.3** | 23.46±0.09 | **2.38**±0.12 | 11.58±0.13 | 34.67±0.24 |

**Table B.3.** Comparison of the relative training speed, relative prediction speed, error rate, ECE, NLL, and Brier on CIFAR-100. For each metric, the lower value means the better. Relative training and relative prediction times are normalized with respect to the baseline method. Best results are indicated in bold.

| Data | Architecture | Method | Relative Training Time | Relative Prediction Time | Error Rate(%) | ECE(%) | NLL(%) | Brier Score(%) |
|---|---|---|---|---|---|---|---|---|
| CIFAR-10 | ResNet-110 | Baseline | 1.0 | 1.0 | 5.89 | 4.46 | 3.34 | 10.2 |
| | | MCDrop | 1.0 | 5.0 | 5.93 | 3.96 | 2.57 | 9.7 |
| | | DeepEnsemble | 9.5 | 5.0 | **5.44** | 5.72 | **2.43** | **8.81** |
| | | NeuBoots | **1.1** | **1.2** | 5.65 | **0.89** | 3.28 | 9.32 |
| | DenseNet-100 | Baseline | 1.0 | 1.0 | 5.13 | 3.2 | 2.23 | 8.3 |
| | | MCDrop | 1.1 | 5.0 | 4.95 | 2.72 | 1.93 | 8.1 |
| | | DeepEnsemble | 9.5 | 5.0 | 4.63 | **0.54** | **1.46** | **6.74** |
| | | NeuBoots | **1.1** | **1.3** | **4.0** | 2.87 | 2.82 | 8.66 |
| SVHN | ResNet-110 | Baseline | 1.0 | 1.0 | 3.55 | 2.39 | 1.75 | 5.8 |
| | | MCDrop | 1.1 | 5.0 | 3.64 | 1.8 | 1.73 | 6.11 |
| | | DeepEnsemble | 9.5 | 5.0 | **2.65** | 1.78 | **1.2** | **4.16** |
| | | NeuBoots | **1.1** | **1.2** | 3.51 | **0.96** | 1.48 | 5.6 |
| | DenseNet-100 | Baseline | 1.0 | 1.0 | 3.6 | 3.2 | 2.23 | 8.3 |
| | | MCDrop | 1.1 | 5.0 | 3.6 | 1.6 | 1.62 | 5.89 |
| | | DeepEnsemble | 9.5 | 5.0 | **2.68** | 1.55 | **1.18** | **4.23** |
| | | NeuBoots | **1.1** | **1.3** | 3.65 | **0.47** | 1.49 | 5.7 |

**Table B.4.** Comparison of the relative training speed, relative prediction speed, error rate, ECE, NLL, and Brier on various datasets and architectures. For each metric, the lower value means the better. Relative training and relative prediction times are normalized with respect to the baseline method. Best results are indicated in bold.

## B.4  Dropout vs NeuBoots

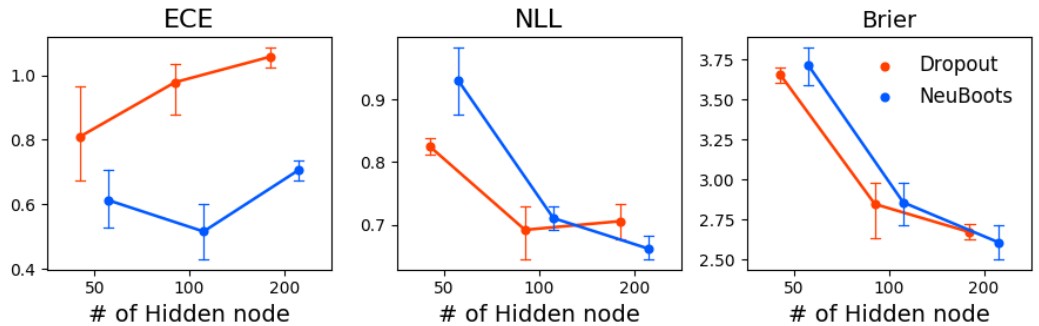

**Figure. B.3.** Comparison of ECE, NLL, and Brier for Dropout and the NeuBoots on the classification task on MNIST.

## B.5  Calibration on Corrupted Dataset

For evaluating the calibrated prediction of NeuBoots under distributional shift, we use Corrupted CIFAR-10 and 100 datasets [18]. Based on Ovadia et al.[33], we first train the ResNet-110 models on each training dataset of CIFAR-10 and 100, and evaluate it on the corrupted dataset. For evaluation, we measure the mean accuracy and standard deviation for each of the five severities.

Deep Ensemble perform best in most cases, but NeuBoots also show better accuracy than baseline.

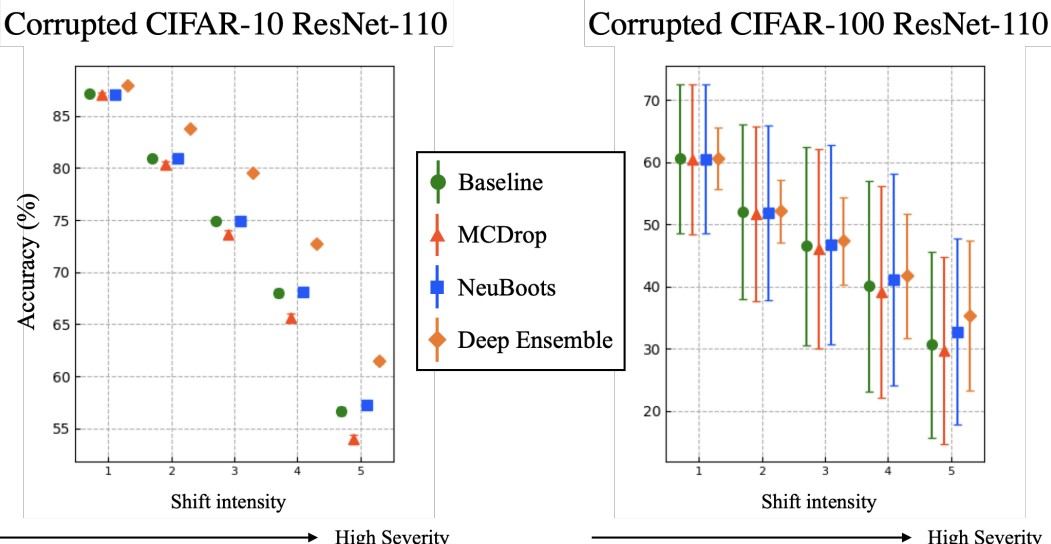

**Figure. B.4.** Calibration under distributional shift.

## B.6  OOD Detection

In this section, we illustrate additional results of OOD detection experiments.

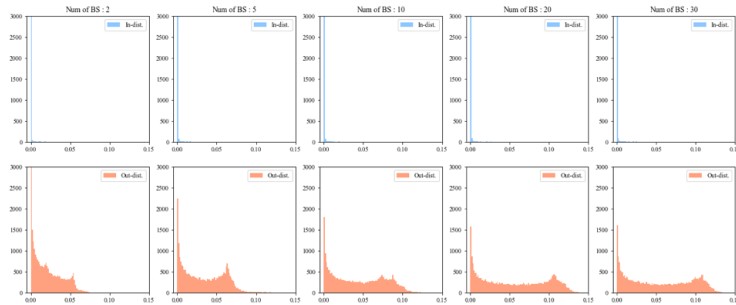

**Figure. B.5.** Histogram of the predictive standard deviation estimated by NeuBoots on test samples from CIFAR-10 (in-distribution) classes (top row) and SVHN (out-distribution) classes (bottom row), as we vary bootstrap sample size $B \in \{2, 5, 10, 20, 30\}$.

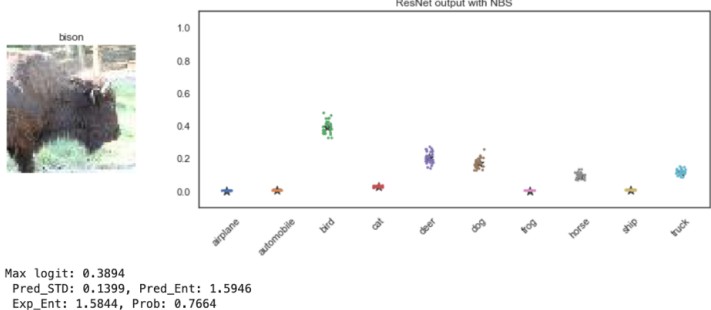

**Figure. B.6.** Confidence bands of the prediction of NeuBoots for `bison` data in TinyImageNet. The proposed method predicts is as an out-of-distribution class with `prob=0.7664`.

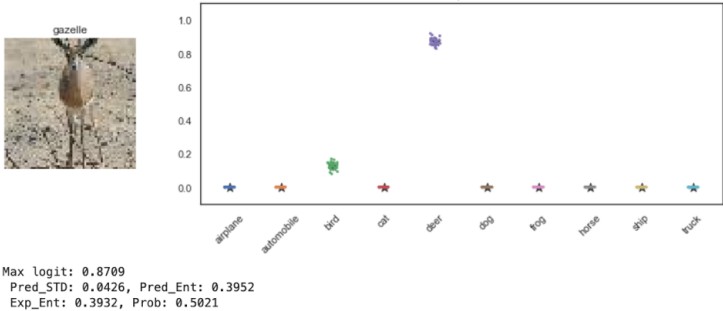

**Figure. B.7.** Confidence bands of the prediction of NeuBoots for `gazelle` data in TinyImageNet. The proposed method predicts is as an out-of-distribution class with `prob=0.5021`.

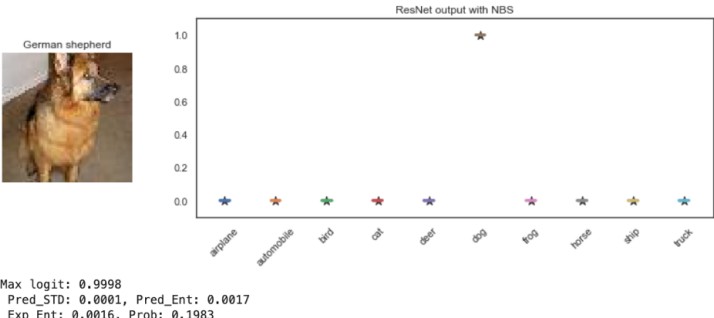

**Figure. B.8.** Confidence bands of the prediction of NeuBoots for `German shepherd` data in TinyImageNet. The proposed method predicts is as an in-of-distribution class `dog` with `prob=0.1983`.