# OpenReview forum: "Neural Bootstrapper"
_NeurIPS.cc/2021/Conference — NeurIPS 2021 Poster_

### Official Review · Reviewer_P38k · 2021-07-16

**Rating:** 7
**Confidence:** 4

**Summary:**

The paper leverages ideas from statistical bootstrapping and proposes a neural network architecture and training loss that can computationally efficiently generate uncertainty estimates. The method is evaluated on a wide range of tasks where uncertainty quantification is important and shows comparable or better performance compared to some popular methods.

**Limitations And Societal Impact:**

Not discussed.

**Main Review:**

The paper proposes a practical approach for uncertainty quantification and conducts an extensive set of experiments that illustrate the effectiveness of the approach. I think this paper will add value to the community.

While the experiments are quite extensive, I think the paper will benefit a lot from providing more justification/explanation for the proposed approach.

For example, the authors might want to comment on the following sanity check: given infinite data, is the resulting distribution going to concentrate on the right function?

Also, the index function that assigns an integer index to each data point is not very well motivated. It feels strange to relate a datapoint to a particular node in the last layer. Why not make the mapping smoother? Instead of projecting alpha on one-hots, why not project on random unit vectors, etc.?

Other comments:
- Loss (2.3): When the weights are all positive, wouldn’t this loss lead to the same minimizer over all possible weights when the network is overparameterized, and then there is no diversity? (Although this doesn’t apply to the architecture you propose.)
- Line 66 Bootstrap Distribution Generator: The idea seems very similar to [1] (although [1] considers a particular form of additive bootstrap in the loss, it can be easily extended to other forms of bootstrapping). Could you comment on the difference.
- Section 4.2 Active Learning: Why not compare to ensembles?

[1] Hypermodels for Exploration, ICLR 2020.


**Time Spent Reviewing:**

3

---

> ### Author Response · Authors · 2021-08-09
> **Response to Reviewer 3(P38k)**
>
>
> **Q: For example, the authors might want to comment on the following sanity check: given infinite data, is the resulting distribution going to concentrate on the right function?**
>
> A: Thank you for the insightful point. Theoretically, bootstrap predictions by the NeuBoots should be equivalent to the standard bootstrap predictions (you can find the theoretical result in the appendix). Of course, as you raised below this result may not be free from the overfitting issue. However, under an ideal setting where the sample size goes to infinity and the neural net structure is fixed, the network is not over-parameterized due to the infinite sample size. In that setting, our theory shows that the resulting bootstrap prediction of NeuBoots is matched with the target prediction from the standard bootstrap.
>
> ---
>
> **Q: Also, the index function that assigns an integer index to each data point is not very well motivated. It feels strange to relate a data point to a particular node in the last layer. Why not make the mapping smoother? Instead of projecting alpha on one-hots, why not project on random unit vectors, etc.?**
>
> A: Thank you for the suggestion. The role of the index function is to assign data points to chosen blocks a priori. In the loss function, each bootstrap weight controls the effect of samples in the corresponding block; a larger value of weight indicates more prediction influence of the data points in the corresponding block. If we randomly perturb all samples, which is equivalent to consider projecting the weight to randomly selected samples during training, the connection between the bootstrap weights and pre-assigned samples will be eliminated. Then, the trained minimizer of the loss function would not be an optimal generator.
>
> ---
>
> **Q: Loss (2.3): When the weights are all positive, wouldn’t this loss lead to the same minimizer over all possible weights when the network is overparameterized, and then there is no diversity? (Although this doesn’t apply to the architecture you propose.)**
>
> A: Thank you for raising this point. Over-parameterization would reduce the variability of the bootstrap distribution, but the element-wise multiplication layer in the architecture alleviates the risk of over-fitting during the training. We think that this architecture has some regularization effects as in dropout during training. Of course, dropouts are strictly different from our NeuBoots in that unlike dropouts, the random weights involve the training loss of the NeuBoots. Our empirical results also show that the over-parameterization issue does not degrade the performance of NeuBoots.
>
> ---
>
> **Q: Line 66 Bootstrap Distribution Generator: The idea seems very similar to [1] (although [1] considers a particular form of additive bootstrap in the loss, it can be easily extended to other forms of bootstrapping). Could you comment on the difference.**
>
> A:  We were not aware of the existence of this paper. Thank you for the reference. The hyper model looks similar with our NeuBoots in that both use independent random variables to generate uncertain predictions. However, NeuBoots aims at generating bootstrap samples, and our theory (in the appendix) shows that under some regularity conditions, NeuBoots achieves valid bootstrap samples when the optimization ideally goes well. In contrast, the hyper model is less related to bootstrapping, and it measures uncertainty of models by looking at the disparity of models from a base model.
>
> ---
>
> **Q: Section 4.2 Active Learning: Why not compare to ensembles?**
>
> A: Initially, we excluded the DE method in the active learning task since their calibration performance was not competitive in CIFAR-100. We have conducted new experiments for DeepEnsembles for additional information and will be added to the revised paper.

---

> > ### Comment · Reviewer_P38k · 2021-08-25
> > **Thanks for the response**
> >
> > Thank you for your response.
> >
> > Some followup clarification questions/comments.
> > > Q: Also, the index function that assigns an integer index to each data point is not very well motivated. It feels strange to relate a data point to a particular node in the last layer. Why not make the mapping smoother? Instead of projecting alpha on one-hots, why not project on random unit vectors, etc.?
> >
> > > A: Thank you for the suggestion. The role of the index function is to assign data points to chosen blocks a priori. In the loss function, each bootstrap weight controls the effect of samples in the corresponding block; a larger value of weight indicates more prediction influence of the data points in the corresponding block. If we randomly perturb all samples, which is equivalent to consider projecting the weight to randomly selected samples during training, the connection between the bootstrap weights and pre-assigned samples will be eliminated. Then, the trained minimizer of the loss function would not be an optimal generator.
> >
> > I didn't mean perturbing all the samples. Currently the bootstrap weights are calculated by projecting alpha down to a single coordinate determined by some pre-determined function u. Why not make this projection smoother? I don't see a good motivation for letting the importance of a (block of) data point affect the value of a particular node in the last layer.
> >
> > > Q: Line 66 Bootstrap Distribution Generator: The idea seems very similar to [1] (although [1] considers a particular form of additive bootstrap in the loss, it can be easily extended to other forms of bootstrapping). Could you comment on the difference.
> >
> > > A: We were not aware of the existence of this paper. Thank you for the reference. The hyper model looks similar with our NeuBoots in that both use independent random variables to generate uncertain predictions. However, NeuBoots aims at generating bootstrap samples, and our theory (in the appendix) shows that under some regularity conditions, NeuBoots achieves valid bootstrap samples when the optimization ideally goes well. In contrast, the hyper model is less related to bootstrapping, and it measures uncertainty of models by looking at the disparity of models from a base model.
> >
> > In that paper, the additive data perturbation is also a form of bootstrapping - just wanted to point out the similarity.

---

> > > ### Author Response · Authors · 2021-08-29
> > > **RE: Thanks for the response**
> > >
> > > We appreciate the reviewer's thoughtful response and suggestion. We leave the additional response regarding the reviewer's question.
> > >
> > > > Why not make this projection smoother? I don't see a good motivation for letting the importance of a (block of) data point affect the value of a particular node in the last layer
> > >
> > > - First of all, the authors admit that we unintentionally omit the motivation of index function and adaptive block bootstrap in subsection 3.1. We will supplement the below explanation in the updated paper.
> > > - The main motivation of the adaptive block bootstrap is to make each node in the last layer learn the features of sampled data corresponding to bootstrap weight.
> > > Suppose $\alpha = (1,...1)$, then each node in the last layer would learn features from uniformly given data as usual. On the other hand, if $\alpha = n \times (1,0,...,0)$, every node becomes zero except for the first node, and the survived node encodes the features of the corresponding block of data. In this way, NeuBoots trains $F_{\theta_{1}}$ to be weak learners by the gradient flow passing each node in the last layer, controlled by bootstrap weights. Even though this is an extreme case and does not happen in RWB, it provides an intuition to understand the mechanism of adaptive block bootstrap.
> > > - The suggested smoothing idea may extend the class of weak learners. We will consider the reviewer's suggestion in future research to bring advances in the proposed method.
> > >
> > > ---
> > >
> > > > In that paper, the additive data perturbation is also a form of bootstrapping - just wanted to point out the similarity.
> > >
> > > - Thanks for the point that we overlooked. Indeed, the hyper model looks similar to a parametric bootstrapping assuming the parametric variance is one. We will cite this reference in the revised paper.

---

> > > > ### Comment · Reviewer_P38k · 2021-08-30
> > > > **Thanks**
> > > >
> > > > Thanks for the clarification.

---

### Official Review · Reviewer_BttC · 2021-07-17

**Rating:** 3
**Confidence:** 3

**Summary:**

The work proposes NeuBoots: a new ensembling method for neural networks inspired by bootstrapping estimation. The method is simple and involves training one neural network to simultaneously represent a whole family of models; each such model is scored during training with a loss where different training objects have different weights. These models are obtained from the base model with a single multiplication of weights of training objects (or, more precisely, weights of sets of training objects) and representations in the final part of the network. During testing, models are ensembled for multiple random weightings for training objects. The authors validate the method on computer vision tasks and synthetic data. It is concluded that the method is superior to other bagging methods on various metrics related to uncertainty estimation such as active learning performance or out-of-distribution detection metrics and is more computationally efficient.

**Limitations And Societal Impact:**

Limitations of the method don't seem to be discussed at length.

**Main Review:**

The method and the experiments are explained with decent clarity. I like the overall composition of the paper, but at a finer level there are some issues. Some details in the narrative are unclear. For example, I didn’t understand why GBS performance degrades and there isn’t data on this in the paper to see for myself. Also, when looking at algorithm 1 on page 4 and the related description, I didn’t understand if sampling of alpha is occurring every optimization step or every epoch; the text states that it’s only happening once per epoch, but then this means that the final network only works for one alpha vector and isn’t trained to work for any other alpha for the whole last epoch? Shouldn’t that lead to deterioration of ensembling performance?

The method mostly builds on the GBS bootstrapping method. NeuBoots reduces computational complexity of bootstrapping very nicely, but also it isn’t clear why elementwise multiplication of block bootstrap weights with representations in a neural network was chosen, compared to any other operation, and this isn’t touched upon in the paper.

Finally, there are linguistic errors (>3 per page on average).

The experiments are sufficiently diverse and comparisons of the method are presented with other method on various tasks and with various architectures and datasets. The results on some tasks like out-of-distribution detection and semantic segmentation seem good. However, parts of the evaluation were confusing for me and I think there are some issues with it.

In Table 3.1 it is shown that NeuBoots outperforms the traditional bootstrap significantly with equal numbers of ensembling members. It isn’t immediately clear to me why this does occur, because as I understand it NeuBoots only presents an approximation to the standard bootstrap with various things relaxed (amortization, weights for blocks of training set examples instead of for individual training set examples, “soft” weights versus binary “hard” weights). Of course it’s completely possible that it just happens to work better for neural models in general but authors don’t attempt to provide an explanation for this in the text.

The paper doesn’t present a comparison of NeuBoots and various methods which, in my opinion, actually make state-of-the-art trade-offs between uncertainty estimation quality and computational complexity of training and inference. Among such methods I can name [MIMO](https://openreview.net/forum?id=OGg9XnKxFAH), [BatchEnsemble](https://arxiv.org/abs/2002.06715), [rank-1 BNNs](http://proceedings.mlr.press/v119/dusenberry20a.html) (see the comparison here: https://github.com/google/uncertainty-baselines/tree/main/baselines/cifar). Instead, it is most often compared with MC dropout in the paper, which is not a state-of-the-art method. Based on the values that I see in those papers it seems like they would provide better uncertainty estimation performance as measured at least by some metrics (e.g. NLL) than NeuBoots. Because of this, it’s difficult to establish the significance of NeuBoots.

The same concern about absence of comparison to state-of-the-art may apply to out-of-distribution detection as well, see, e.g. [Energy-based Out-of-distribution Detection](https://arxiv.org/abs/2010.03759).

The authors may answer that bagging is orthogonal to other ensembling methods and should mostly be evaluated separately. I think I would not agree with this as it isn't clear NeuBoots would yield benefits when combined and applied together with state-of-the-art ensembling methods. See for example [Why M Heads are Better than One: Training a Diverse Ensemble of Deep Networks](https://arxiv.org/abs/1511.06314) where they showed that combining bootstrap with Deep Ensembles can reduce classifier accuracy.

The comparison between methods in Table 4.1 is confusing. NeuBoots is compared with Deep Ensemble, Monte-Carlo dropout, and usual neural network training there.
- The major source of confusion for me here is that the performance of the Deep Ensemble method with 5 networks in the ensemble is worse than a single neural network training. In principle prior work established it to be much better, see, e.g. analogous benchmark comparison in Ashukha et al: [Pitfalls of In-Domain Uncertainty Estimation and Ensembling in Deep Learning](https://arxiv.org/pdf/2002.06470.pdf), Table 3, row 6.
- The text mentions that adversarial training and the Brier loss function is utilized for Deep Ensemble, but presumably not for other methods, which may be seen as unfair, because most often Deep Ensemble is evaluated without these additions (see e.g. in MIMO, BatchEnsemble, rank-1 BNNs papers with links above).
- ECE, NLL and Brier Score metrics are given in percents, but it isn’t clear what is taken as 100% on each metric.
- It seems from Table 4.1 that NeuBoots ensemble NLL is worse compared to a single network. However, it is shown that NeuBoots outperforms a single network and MC dropout on tasks like semantic segmentation and imbalanced classification. This discrepancy in performance on various methods is a bit surprising and it isn’t explained by the authors.


**Time Spent Reviewing:**

5

---

> ### Author Response · Authors · 2021-08-10
> **Response to Reviewer 2(BttC)**
>
> **Q: I didn’t understand why GBS performance degrades and there isn’t data on this in the paper to see for myself. Also, when looking at algorithm 1 on page 4 and the related description, I didn’t understand if sampling of alpha is occurring every optimization step or every epoch; the text states that it’s only happening once per epoch, but then this means that the final network only works for one alpha vector and isn’t trained to work for any other alpha for the whole last epoch? Shouldn’t that lead to deterioration of ensembling performance? The method mostly builds on the GBS bootstrapping method. NeuBoots reduces the computational complexity of bootstrapping very nicely, but also it isn’t clear why elementwise multiplication of block bootstrap weights with representations in a neural network was chosen, compared to any other operation, and this isn’t touched upon in the paper.**
>
> A: Thanks for your comments. Theoretically, GBS and NeuBoots should be equivalent to the standard bootstrap predictions. Over-parameterization would reduce the variability of the bootstrap distribution so that GBS can be prone to overfitting on the bootstrap weights at the final epochs by vanishing coefficients corresponding to concatenated bootstrap weights, and this could be a possible factor why GBS does not work practically for the neural networks. However, for NeuBoots, element-wise multiplication in the final layer alleviates the risk of over-fitting during the training. We think empirical results on imbalanced data and OOD support that the overfitting does not occur during the NeuBoots training. We observe that this structure has stochastic regularization effects as in dropout during training (see NeuBoots vs Dropout in Section 3.2).
>
> ---
>
> **Q: In Table 3.1 it is shown that NeuBoots outperforms the traditional bootstrap significantly with equal numbers of ensembling members. It isn’t immediately clear to me why this does occur, because as I understand it NeuBoots only presents an approximation to the standard bootstrap with various things relaxed (amortization, weights for blocks of training set examples instead of for individual training set examples, “soft” weights versus binary “hard” weights). Of course it’s completely possible that it just happens to work better for neural models in general but authors don’t attempt to provide an explanation for this in the text.**
>
> A: We appreciate your scientific comments. Again, we would like to point out that naive bootstrapping neural networks are challenging due to their nature of repetitive sampling and training, as we mentioned in the Introduction. As the Reviewer mentioned, NeuBoots is an approximation method; however, its training is more efficient than the standard bootstrapping (see Figure 3.3). We will explain this result in the corresponding part.
>
> ---
>
> **Q: The major source of confusion for me here is that the performance of the Deep Ensemble method with 5 networks in the ensemble is worse than a single neural network training. In principle prior work established it to be much better, see, e.g. analogous benchmark comparison in Ashukha et al: Pitfalls of In-Domain Uncertainty Estimation and Ensembling in Deep Learning, Table 3, row 6.**
>
> ---
>
> A: Thanks for your kind comment. We found there is some discrepancy between [Ashukha et al](https://openreview.net/pdf?id=BJxI5gHKDr) and our experimental setting. Importantly, we did not use augmentation, which can be a dominating factor in CIFAR-100, to measure the pure uncertainty estimation performance of NeuBoots. We checked that the overall performance in Table 4.1 can be improved if we follow the training setting in [Ashukha et al](https://openreview.net/pdf?id=BJxI5gHKDr). We will mention this part clearly in the revised paper.
>
> ---
>
> For the other questions, please see the [Common Response](https://openreview.net/forum?id=Hk2oOy4GJlH&noteId=HFEHgtKwYPE) above. We compared the proposed method with MIMO and BatchEnsemble in terms of computational complexity. Also, we supplemented DE results (old and new) in segmentation tasks. The authors believe this would mitigate the Reviewer's concern on the proposed method. Again, we sincerely appreciate the Reviewer's scientific comments.

---

### Official Review · Reviewer_S2ta · 2021-07-20

**Rating:** 7
**Confidence:** 4

**Summary:**

This work advances an approach for generating bootstrapped neural networks through a single training run, instead of training individual networks for each bootstrap subsets of the dataset. The method, dubbed Neural Bootstrapper (NeuBoots), can be seen as a last-layer approach for uncertainty quantification with an adaptive bootstrapping block at the end. This bootstrapping block borrows ideas from random weight bootstrapping, generative bootstrap samplers and block bootstrapping. In detail, groups of samples are given the same bootstrapping weight (sampled from a Dirichlet distribution randomly initialized at each epoch), with this weight used for perturbing the penultimate layer features and to ponderate the loss for each corresponding sample. Multiple predictions can be obtained from different samplings of the bootstrap weights applied again to the penultimate layer features leading to computationally efficient bagging.
NeuBoots is evaluated on an array of datasets against MC-Dropout and DeepEnsembles, displaying interesting results.


**Limitations And Societal Impact:**

Limitations and societal impact were not discussed.


**Main Review:**


## Recommendation
I find that this paper advances a nice idea with lots of experiments and results. However a large body of literature is missing from this discussion and evaluation (NeuBoots is essentially compared against baseline and MC-Dropout and on image classification against DeepEnsemble), while some of the inner workings of NeuBoots are not clear.
I'm currently leaning towards rejection. Please check more detailed notes below and at the end some questions for improving the paper or for the rebuttal.


## Paper strengths
- in essence, I find the idea of this paper really nice. Directly adapting bootstrapping to neural networks hasn't lead to exciting results so far [a] and this work advances an interesting approach

- the method is simple (in spite of the loaded formalism used for presenting the concept) and relatively accessible to deploy. The authors have readily shared the implemetation code.

- NeuBoots is evaluated over a rich range of settings and datasets. Few uncertainty estimation works venture to this amount of tasks.


## Paper weaknesses

### Related work
- This section ignores a large part of the literature on uncertainty quantification in deep learning from the past years, essentially stopping at MC-Dropout and DeepEnsemble. There has been significant activity in last-layer (Prior Networks [b], evidential models [c], multiple last-layer approaches [e], DUQ [f], etc.) and/or single training (SWAG [g], TRADI [h], PEP [i]), efficient ensembles (BatchEnsemble [j]) research that is not mentioned nor evaluated here.

- I recommend including some of these methods for discussion and comparison

- please note that [e] also used bagging on the last layer of a previously trained network, though the method was not as advanced as NeuBoots.

### Quality and clarity
- I appreciate that the paper and the appendix are rich in experiments and proofs towards better assessing the benefits of this methods. However, to my understanding, there are a few aspects of this method that are not clear from the provided description and intuition for NeuBoots. See below:

- **Diversity.** The bootstrap block weights $\mathbf{w}_{\alpha}$ are randomly initialized at each epoch from $\boldsymbol \alpha$ that is sampled Dirichlet distribution. If I understand correctly $\boldsymbol \alpha$ is never updated. The intuition is that at each epoch the network is trained on a different bootstrap subset. As networks are known to be prone for catastrophic forgetting, there is a risk that the network to "overfit" on the bootrap from the last epoch(s) and forget the previous ones. From what I can see there is no explicit regularization mechanism in NeuBoots to deal with that. One way to assess this is to analyze the diversity of the predictions (diversity is usually one of the main advantages of ensembles [k],[j]): a higher diversity would infirm this hypothesis. Usually more compelling diversity scores are obtained on distribution shift (e.g., ImageNet-C, Cifar-C [l]) or OOD. The confidence bands in OOD plots in the appendix do not hint a high diversity.

- **Grouping samples in blocks**. Block boostrapping has been mostly used for correlated or redundant samples. In this case, the samples are seemingly grouped from a random initialization before training. Did the authors try other strategies? Was there an impact? It would be useful to include some strategies or rules of thumb for selecting the number of segments S. In the active learning experiments, it is not clear how are the new samples added to the bootstrap segments.


### Experiments
- as mentioned above, there are several other efficient uncertainty approches that are missing from the discussion and evaluation. MCDrop and DeepEnsemble are certainly useful baselines, but there are many works in-between and it would be helpful to know where is NeuBoots placed.

- accuracy and calibration on in-domain data offers just a partial view on the effectiveness of the approach. CIFAR experiments can be extended to dataset shift [m],[l], where uncertainty estimation and calibration is even more important.

- for the OOD experiments I believe there are some inconsistencies in the description: in L206 "we tune hyperparameters in the training phase using in-distribution samples to keep the fairness of our method", then in L213 "For our method, we tune whole hyperparameters using a separate validation set, which consists of 1,000 images from in-distribution and out-distribution, respectively". It's not clear whether NeuBoots uses the OOD data for training or not. Also for OOD detection, the classifier prediction values can be used directly without the requirement of training a logistic regression for this.
Please note that most of the recent approaches refrain from using OOD during training as it's choice can largely impact the performance on the test OOD set [n]. Directly evaluating on unknown OOD data (as done in DeepEnsembles [18])


## Questions
Here are some questions that could be potentially addressed in the rebuttal:

1. How is NeuBoots doing in terms of diversity of predictions? Can the authors please explain if and how NeuBoots does not "overfit" on the bootstrap weights from the final epochs?

2. I suggest a simple baseline to validate that NeuBoots had learned richer information and can come up with diversity in predictions. This baseline consists in leveraging Gaussian Dropout (essentially perturbing the hidden features with noise sampled from a user-defined Guassian, section 10 in [o]) to perturb last layer features and generate and ensemble like NeuBoots does. How is NeuBoots expected to perform here?

3. How is NeuBoots expected to perform against other computationally efficient methods for uncertainty estimation?

4. Is there any recommended strategy for grouping samples in blocks?


**References:**


[a] J. Nixon et al., Why Are Bootstrapped Deep Ensembles Not Better?, NeurIPS 2020 workshop

[b] A. Malinin and M. Gales, Predictive uncertainty estimation via prior networks, NeurIPS 2020

[c] M. Sensoy et al., Evidential deep learning to quantify classification uncertainty, NeurIPS 2018

[d] T. Joo et al., Being bayesian about categorical probability, ICML 2020

[e] N. Brosse et al., On Last-Layer Algorithms for Classification: Decoupling Representation from Uncertainty Estimation, arXiv 2020

[f] J. van Amersfoort et al., Uncertainty estimation using a single deep deterministic neural network, ICML 2020

[g] W. Maddox et al., A Simple Baseline for Bayesian Uncertainty in Deep Learning, NeurIPS 2019

[h] G. Franchi et al., TRADI: Tracking deep neural network weight distributions, ECCV 2020

[i] A. Mehrtash et al., PEP: Parameter Ensembling by Perturbation, NeurIPS 2020

[j] Y. Wen et al., BatchEnsemble: An Alternative Approach to Efficient Ensemble and Lifelong Learning, ICLR 2020

[k] S. Fort et al., Deep Ensembles: A Loss Landscape Perspective, arXiv 2019

[l] D. Hendrycks and T. Dietterich,  Benchmarking Neural Network Robustness to Common Corruptions and Perturbations, ICLR 2019

[m] Y. Ovadia et al., Can You Trust Your Model's Uncertainty? Evaluating Predictive Uncertainty Under Dataset Shift, NeurIPS 2020

[n] A. Shafaei, A Less Biased Evaluation of Out-of-distribution Sample Detectors, BMVC 2019

[o] N. Srivastava et al., Dropout: A Simple Way to Prevent Neural Networks from Overfitting, JMLR 2014


--------------------------------------------------
--------------------------------------------------

## Final recommendation

The authors have addressed in the rebuttal several concerns from reviewers and I thank them for the detailed responses, clarifications and new experiments. They are not wasted and I'm sure they'll contribute to making the paper stronger.

After the exchange of questions and responses with the authors, reading other reviews and responses and the paper again I would recommend this paper for acceptance.
The paper advances an interesting idea and the additional experiments and clarifications should enforce the findings in this work, i.e., that there is some form of bootstrapping happening.
However, I find that the experimental evaluation is not complete yet. As mentioned in the discussions, I strongly encourage the authors to evaluate NeuBoots on data distribution shift (CIFAR-C) and to include other relevant and relatively recent methods, i.e., newer than MC-Dropout,  in their comparison.


**Time Spent Reviewing:**

5

---

> ### Author Response · Authors · 2021-08-10
> **Response to Reviewer 1(S2ta)**
>
>
> **Related Work**
>
> We sincerely appreciate the Reviewer's suggestion. We will cite the suggested works in the revised paper.
>
> **OOD Experiment Setting**
>
> We thank the Reviewer for pointing out the ambiguity of our statement.
> Our setting for OOD is based on the baseline model[1].
> 1. Train a classifier only using the training set of the In-distribution dataset (50000 samples for CIFAR-10).
> 2. Evaluate all test datasets, both In and Out (10000 samples for CIFAR-10 and 10000 samples for SVHN).
> 3. Extract four features; maximum score, the max of predictive mean vectors, the standard deviation of logit vectors, expected entropy, and predictive entropy.
>
> Due to the nature of multiple features, we train the logistic regressor instead of picking the thresholds.
> This is the only difference with the baseline, which selects a particular threshold as a hyperparameter.
> For training the logistic regressor(or selecting the threshold), we split the test dataset into two groups; training(10% of test set) and evaluation(90% of test set), same as the baseline[1], ODIN[2], Mahalobis[3], and energy-based OOD detection[4].
>
> Thus, the setting of our method is exactly the same as the OOD of Deep Ensemble[5]; that is what you expect.
>
> References:
> [1] D. Hendrycks and K. Gimpel, A baseline for detecting misclassified and out-of-distribution examples in neural networks
> [2] S. Liang et al., Enhancing the reliability of out-of-distribution image detection in neural networks
> [3] K. Lee et al., A simple unified framework for detecting out-of-distribution samples and adversarial attacks
> [4] W. Liu et al., Energy-based out-of-distribution detection
> [5] B. Lakshminarayanan et al., Simple and scalable predictive uncertainty estimation using deep ensembles
>
>
>
> ---
>
> Below are answers to raised questions from the Reviewer.
>
> **Q: How is NeuBoots doing in terms of diversity of predictions? Can the authors please explain if and how NeuBoots does not overfit on the bootstrap weights from the final epochs?**
>
> A: Thanks for the insightful comment. Note that we sample $\mathbf{\alpha}$ from Dirichlet distribution; hence it has essentially positive weights for each data point, as we already explained in the RWB section. Hence for each epoch, the network observes every data point and this situation is substantially different from catastrophic forgetting. GBS can be prone to overfitting on the bootstrap weights at the final epochs by vanishing coefficients corresponding to concatenated bootstrap weights, and this could be a possible factor why GBS does not work practically for the neural networks. However, for NeuBoots, element-wise multiplication in the final layer alleviates the risk of over-fitting during the training. We think empirical results on imbalanced data and OOD support that the overfitting does not occur during the NeuBoots training.
>
> ---
>
> **Q: I suggest a simple baseline to validate that NeuBoots had learned richer information and can come up with diversity in predictions. This baseline consists in leveraging Gaussian Dropout (essentially perturbing the hidden features with noise sampled from a user-defined Guassian, section 10 in [o]) to perturb last layer features and generate and ensemble like NeuBoots does. How is NeuBoots expected to perform here?**
>
> A: Thanks for the suggestion. Indeed, we compared NeuBoots to a similar setting (see NeuBoots vs Dropout paragraph in Section 3.2). We have applied Dropout only to the last layer features and compared it with NeuBoots. Overall, the bagging performance was similar, but NeuBoots outperforms Dropout in ECE, NLL, and the Brier score. This tendency does not change even we implement the different Dropout settings as the Reviewer suggested. We believe that Random Weight Bootstrapping (RWB) is a crucial factor in this result. Please note that RWB based NeuBoots also outperform standard bootstrap and amortized bootstrap in various settings (see NeuBoots vs Amortized Bootstrapping paragraph in Section 3.2).
>
> ---
>
> **Q: How is NeuBoots expected to perform against other computationally efficient methods for uncertainty estimation?**
>
> A: As mentioned above, we compared NeuBoots to MIMO and BatchEnsemble in terms of training, test, and memory costs. All methods are comparable in test time and have computational efficiency in the usual image classification task. However, MIMO and BatchEnsemble require batch repetition in the training time; hence their training becomes longer as the number of ensembles increases. Also, if the input data is high-dimensional, they require more memories (see the first table in Common Response). We also emphasize that NeuBoots outperforms DeepEnsemble in the segmentation tasks without losing computational efficiency, although MIMO and BatchEnsemble did not demonstrate their uncertainty estimation in segmentation tasks.
>
> ---
>
> **Q: Is there any recommended strategy for grouping samples in blocks?**
>
> A: Thank you for asking this point. We examined multiple randomized groupings, and the results were consistent across different groupings. We also considered a grouping strategy based on stratified sampling, but the results were almost the same. We believe that as long as there exist no extreme outliers, the effect from the grouping would be minimal. We also concern situations where a different grouping influences the bootstrap results due to the existence of outliers or some other reasons. To avoid this, we recommend comparing the bootstrap distributions from different groupings, if the computational resource is allowed.

---

> > ### Comment · Reviewer_S2ta · 2021-08-17
> > **Comments and clarification after rebuttal**
> >
> >
> > I would like to thank the authors for their dense responses to my questions as well as concerns and questions from other reviewers. I appreciate them.
> >
> > I would still have a few comments and potential misunderstandings (from my side or the authors').  I will regroup my comments and questions as in the response from the authors.
> >
> > **OOD Experiment Setting**
> > - as expressed in my initial concern, the OOD experiments make use of OOD data to train the logistic classifier to separate in-distribution from out-of-distribution. This practice is used less and less as a classifier practically learns the given training OOD dataset and does not guarantee robustness to other OOD datasets.  Shafaei et al. [n] have actually showed how the performance on a certain OOD dataset depends on what OOD dataset was used during training.
> > - the mentioned methods [1-4] do use some type of OOD data during training in way or another: ODIN [2] does use a validation OOD dataset to tune the thresholds, Mahalanobis [3] has two variants (one like ODIN and another one independent of OOD datasets that is tuned with adversarial attacked images -- the latter performs a bit worse), energy OOD [4] uses a random unlabeled dataset for OOD to avoid dependence on a specific OOD dataset. While baseline [1] has a variant that uses an OOD dataset (_Section 4 - Abormality detection with auxiliary decoders_), please not that it's most common usage and utility (across most papers) is to take the maximum softmax prediction as indicator for confidence. The trend is to be as independent as possible from any OOD dataset during training or validation.
> > - I don't see how is the considered setup similar with DE which does not use any tuning on a validation OOD dataset. The maximum prediction score from the average predictions is usually used for most metrics.
> > - I would suggest computing scores like in DE and the original baseline, which is simpler as it does not need definition of features to use, e.g., maximum score, the max of predictive mean vectors, the standard deviation of logit vectors, expected entropy, and predictive entropy, not training another classifier on top of them.
> >
> > --------------
> >
> > **Diversity of predictions**
> > - I can relate to the arguments of the authors but they would be a lot more convincing with some quantitative numbers in terms of diversity.
> > - The block-based grouping might offer some regularization to not over-fit to a certain configuration and weights in the final epoch, but I would be more at ease with some numbers comparing this to MC-Dropout and DE. Diversity is key for uncertainty quantification in unknown conditions and has been so far the appanage of ensemble methods. It would be nice to see how much diversity is attained by the bootrapping strategy.
> > - the authors can find some ideas metrics for measuring diversity (that do not require re-training) in some of these works: ratio-error [p,q], Q-statistics [p,q], correlation coefficient [p,q], prediction disagreement [k]
> >
> > --------------
> >
> > **Gaussian Dropout**
> > - While both the original Dropout and Gaussian Dropout have been originally shown to have similar impact in terms of regularization, I'm not sure that the same impact on calibration, OOD performance, etc., is obvious, in particular when used only over the logits as done here. Standard Dropout sparsifies the activations while keeping magnitude similar with rescaling, while Gaussian Dropout does not impact sparsity. In the last layer only, this can make a difference. Could the authors please elaborate on the implementation and tests used for Gaussian Dropout experiments, please?
> > - The aleatoric uncertainty for classification implementation from Kendall and Gal [r] generates an ensemble of predictions also by perturbing the logits with some noise sampled from Gaussian distribution (in their case the parameters of this distribution are either learned or predicted) (Section 4)
> > - The experiment I suggested was in this line [r], but with a manually tuned noise distribution over an already trained network, e.g., the baseline. Again the aim is to show that the benefits come from the bootstrapping idea and not the test-time augmentation of the logits.
> >
> > To be clear, I'm not aiming to undermine this contribution, but rather to help out in emphasizing that the gains and properties are actually due to the Bootstrapping idea and not from the test-time augmentation of the logits and eventually a fortunate configuration of the hyper-parameters. The goal is to make this paper stronger.
> >
> > -------------------
> > **Other references**
> >
> > [p]  M. Aksela. Comparison of classifier selection methods for improving committee performance, International Workshop on Multiple Classifier Systems 2003
> >
> > [q] A. Rame and M. Cord, DICE: Diversity in Deep Ensembles via Conditional Redundancy Adversarial Estimation, ICLR 2021
> >
> > [r] A. Kendall and Y. Gal, What Uncertainties Do We Need in Bayesian Deep
> > Learning for Computer Vision?, NeurIPS 2017

---

> > > ### Author Response · Authors · 2021-08-20
> > > **Additional response to Reviewer S2ta**
> > >
> > > We greatly appreciate your kind clarification and comments.
> > >
> > > **OOD Detection**
> > >
> > > Thanks for your kind comments. Based on the reviewer's comments, we additionally perform OOD detection using only the *maximum softmax prediction score* in the same way as the baseline without any further training.
> > > We note that only AUROC, AUPR_In, and AUPR_Out can be measured as evaluation metrics.
> > > The other experimental settings are the same as in the main script. (In : CIFAR10, Out : SVHN, Model : ResNet-34)
> > >
> > > |Method|AUROC|AUPR_In|AUPR_Out|
> > > |---|---|---|---|
> > > |NeuBoots (manuscript) |99.14|97.78|99.68|
> > > |NeuBoots (non-training) |98.87|99.52|97.44|
> > >
> > > We reveal that the OOD performance of NeuBoots slightly decreased (except AUPR_In); however, we believe that this is still competitive compared to the other methods in the manuscript, which are not measured by the suggested experimental setting.
> > >
> > > ---
> > >
> > > **Diversity of predictions**
> > >
> > > We greatly appreciate the reviewer's kind suggestion. For the diversity experiments, we measure the four metrics mentioned by the reviewer on DenseNet-100 that performed best in the CIFAR-100 classification task.
> > > A new version of the Deep Ensemble (DE-new) model, which excludes the adversarial training and brier loss, is re-trained and used as a comparison group for a fair comparison and to know our method's relative position.
> > > Although NeuBoots has weaker classification performance compared to DE (23.48% v.s. 20.16% for error rate), it outperforms two (correlation & disagreement) out of four metrics in terms of diversity.
> > >
> > > |Method|Ratio-error($\uparrow$)|Q-Stat$(\downarrow$)|Correlation($\downarrow$)|Disagreement($\uparrow$)|
> > > |---|---|---|---|---|
> > > |DE-new|**98.00**|**61.31**|78.56|23.41|
> > > |MC-Drop|27.38|96.33|92.00|10.40|
> > > |NeuBoots|93.79|63.95|**76.11**|**32.20**|
> > >
> > > We believe that the above result shows bootstrapping strategy can attain diversity comparable to the ensemble methods. This result will be added to the revised manuscript.
> > >
> > > ---
> > >
> > > **Gaussian Dropout**
> > >
> > > Thanks very much for your clarification. We set the control group in two aspects. The first is a method (MC-GaussianDropout) that provides perturbation during training in the same way as NeuBoots, and the second is a method (GaussianDropout-TTA) that performs logit augmentations at test time on a pre-trained baseline model. We use DenseNet on CIFAR-100 and evaluate suggested methods with different variances:
> > >
> > > |Method|Error Rate|ECE|NLL|Brier|
> > > |---|---|---|---|---|
> > > |Baseline|24.02|12.38|10.93|36.40|
> > > |MC-GaussianDropout (variance=0.5)|24.64|11.09|11.86|36.94|
> > > |MC-GaussianDropout (variance=1.0)|25.03|12.08|13.50|38.52|
> > > |GaussianDropout-TTA (variance=0.1)|24.30|9.17|10.00|35.32|
> > > |GaussianDropout-TTA (variance=0.25)|24.30|6.26|10.06|34.50|
> > > |GaussianDropout-TTA (variance=1.0)|27.19|15.62|11.81|42.95|
> > > |MC-Dropout|23.88|10.22|**9.49**|**34.94**|
> > > |NeuBoots|**23.46**|**2.38**|11.58|34.67|
> > >
> > > According to the results, GaussianDropout-TTA seems effective in calibration and uncertainty estimation, unlike MC-GaussianDropout. However, both methods have no performance gain than NeuBoots, which attains better calibration too. Based on the results of the first and second methods, we can clearly confirm the benefits of the bootstrapping of NeuBoots.

---

> > > > ### Comment · Reviewer_S2ta · 2021-08-25
> > > > **Thanks for the responses**
> > > >
> > > > I thank the authors for the thoughtful responses and for the additional results and arguments. They answer my questions in a convincing manner.
> > > >
> > > > For the updated paper, please include DE and MCDropout in the OOD table to preserve consistency with other results across the paper.

---

> > > > ### Comment · Reviewer_S2ta · 2021-08-25
> > > > **Final clarification**
> > > >
> > > > Some final requests from clarification that came up as I was wrapping up the recommendation.
> > > >
> > > > - the authors mentioned a comparison with MIMO and BatchEnsemble, as requested by reviewer _BttC_  (I've asked for a more recent model, newer than MCDropout). There is a cost comparison with these methods in the general response, but there's no quantitative one (the authors seem to hint there is such a comparison if I understand correctly). Could the authors please clarify this? By the way, both MIMO and BatchEnsemble have been used in segmentation tasks (StreetHazards, BDD-Anomaly) as baselines in [s].
> > > >
> > > > - the authors may consider, at least for the updated version, extension of experiments to distribution shift (CIFAR-10/100-C)  [m],[l], for which no re-training is required, just testing on a large set of test images
> > > >
> > > > Thanks!
> > > >
> > > > [s] G. Franchi et al., Encoding the latent posterior of Bayesian Neural Networks for uncertainty quantification, arXiv 2020

---

> > > > > ### Author Response · Authors · 2021-08-28
> > > > > **RE: Final clarification**
> > > > >
> > > > > We appreciate the reviewer's sincere comment and suggestion. For a quantitative comparison between MIMO, BatchEnsemble, and NeuBoots, we additionally conducted simple experiments. We simply estimate training time and prediction time during one epoch and measure the memory cost of those methods (ResNet-110 on CIFAR-100).
> > > > >
> > > > > |K=5|Baseline|NeuBoots|MCDropout|MIMO|BatchEnsemble|DeepEnsemble|
> > > > > |:---:|:---:|:---:|:---:|:---:|:---:|:---:|
> > > > > |training time (seconds)|29.1|29.24|29.24|30.2|217.11|145.5|
> > > > > |inference time (seconds)|1.69|1.71|8.45|1.87|6.59|8.45|
> > > > > |memory cost (mb)|1605|1605|1605|1605|2474|7600|
> > > > >
> > > > > We verified that NeuBoots benefits computational efficiency compared to BatchEnsemble. The memory comparison between NeuBoots and MIMO seems unclear due to the small resolution of CIFAR (we observed NeuBoots has lower memory cost than MIMO if we change the model from ResNet to DenseNet).
> > > > >
> > > > > Interestingly, the experiment shows that the computational cost of MIMO is similar to that of NeuBoots. This result is because we used a convolutional neural network that benefits channel parallelism thanks to GPU. However, if we apply MIMO to MLP, the result is different: for 5-layer MLP (K=3), NeuBoots takes 7.57 seconds for one epoch training, MIMO takes 41.09 seconds, as we expected. Also, we observed that growing the number of the ensemble makes the proportion gap increase: for K=5, NeuBoots takes 7.68 seconds, MIMO takes 62.47 seconds. Furthermore, the below table shows that MIMO requires more memory cost in test time as the number of ensembles increases, as we expected.
> > > > >
> > > > > | |K=1|K=2|K=3|K=4|K=5|
> > > > > |:---:|:---:|:---:|:---:|:---:|:---:|
> > > > > |NeuBoots|1515|1517|1517|1517|1517|
> > > > > |MIMO|1561|1633|1741|1885|2067|
> > > > >
> > > > > The authors believe the above comparison sufficiently clarifies our assertion in general response. The authors will contain the above comparisons in the revised paper. Also, we appreciate introducing the reference [s], we are going to cite it in the Related Work. As the reviewer suggested, the authors are considering a distribution shift experiment on CIFAR-10/100-C in the appendix.
> > > > >
> > > > > Thanks very much again for your scientific comment and discussion.

---

> > > > > > ### Comment · Reviewer_S2ta · 2021-08-30
> > > > > > **Re:**
> > > > > >
> > > > > > Thanks for clarifying this.

---

### Author Response · Authors · 2021-08-09
**Common Response: Thanks for your scientific comments**

The authors greatly appreciate all reviewers' insightful comments. We collect some common questions and leave answers here.

## Comparison with Ensemble methods
Our motivation builds on *bootstrapping*, which is an essential technique in machine learning and statistics. Hence our comparison is mainly focused on traditional bootstrapping and [Amortized Bootstrap](https://www.semanticscholar.org/paper/The-Amortized-Bootstrap-Nalisnick/5c304685be9e6571bffd7dbd77843516cb12f353). Even though bootstrapping and DE shares the same theoretical motivation, DE is essentially a non-bootstrap based ensemble, as [Fort et al.](https://arxiv.org/pdf/1912.02757.pdf) mentioned; hence we initially thought NeuBoots should be evaluated independently of DE's recent variants. Furthermore, we think bootstrapping can be successfully combined with ensemble methods if we can utilize an efficient bootstrap method for deep neural networks, and we claim that the proposed method would be a promising approach to achieve this.

Nevertheless, we admit that comparisons between the proposed method and the recent variants of DE (ex. BatchEnsemble, MIMO) are necessary for two points of view: (a) computational complexity and (b) calibration performance in segmentation task. We are going to supplement the following results in the revised paper to provide additional information for readers.

### (1) Computational Complexity
We focus on the scalable implementation of bootstrapping for deep neural networks, which have been recognized as a challenging task. To compare the proposed method with A and B, we use the following notations: $L$: the number of layers, $K$: the number of bootstrapping (or ensemble), $M$: the number of parameters, $I$: memory size of input data:

- Except for DE, the test complexity of MIMO/BatchEnsemble are the same as one of NeuBoots. However, MIMO/BatchEnsemble require batch repetition in the training time, hence their training complexity are $O(LK)$, whereas NeuBoots has $O(L+K)$ complexity.
- MIMO needs to copy input images as many as $K$ to supply into input layers. Even though it can compute in a single forward pass, if the input data is high-dimensional (ex. MRI/CT), it requires many memories to upload multiple inputs. This computational bottleneck is nothing to sneeze at the aerial or biomedical fields, which requires on-device training or inference. This problem does not happen in NeuBoots since multiple computations occur only at the final layer.
- The above statements also hold between NeuBoots and BatchEnsemble. Even though the memory size of fast weights in BatchEnsemble can be ignorable in almost all classification tasks, it cannot be ignorable in the segmentation task since the memory usage of fast weights in BatchEnsemble is proportional to the dimension of input and output. Consequently, the memory complexity of BatchEnsemble is similar to the one of MIMO, which can be a critical issue in the on-device application. On the other hand, NeuBoots does not require any additional parameters; hence it is free from the above memory problems.

|Method|Training cost|Test cost|Memory cost|
|---|---|---|---|
|DeepEnsemble|$O(LK)$|$O(LK)$|$O(MK+I)$|
|MIMO|$O(LK)$|$O(L+K)$|$O(M+IK)$|
|BatchEnsemble|$O(LK)$|$O(L+K)$|$O(M+IK)$|
|NeuBoots|$O(L+K)$|$O(L+K)$|$O(M+I)$|

### (2) Empirical comparison between NeuBoots and Deep Ensemble (old and new)

We implemented DE training with Brier loss function and adversarial training (DE-old), which were suggested by the original reference [Lakshminarayanan et al.](https://arxiv.org/abs/1612.01474). We thought this setting is a fair comparison; however, following Reviewer 2(BttC)'s comment, we re-experimented DeepEnsemble following the recent studies (DE-new).
DE-new outperforms NeuBoots and the other methods in the image classification experiments. However, we want to clarify that our results are not directly comparable with [Ashukha et al.]([https://openreview.net/pdf?id=BJxI5gHKDr](https://openreview.net/pdf?id=BJxI5gHKDr)) because we did not use augmentation in the experiments to prevent performance overestimation. The overall performance in Table 4.1 can be improved if we follow the training setting in [Ashukha et al.]([https://openreview.net/pdf?id=BJxI5gHKDr](https://openreview.net/pdf?id=BJxI5gHKDr)).

We also conducted additional experiments for the segmentation tasks (PASCAL VOC & NIH3T3) with DE-old and DE-new. Interestingly, both DE methods record better performance than baseline and MCDrop, but NeuBoots remains the best model. Also, NeuBoots almost outperform DeepEnsemble methods in the view of the ECE scores. Note that DE-new records poor performance in the 2D segmentation task. These experiments imply that DE methods sometimes show different performances in distinct tasks beyond CIFAR/ImageNet, and bootstrapping can be another competitive option. We emphasize that NeuBoots is computationally much lighter than DE methods in the segmentation tasks.

1. ResNet-50 on 2D (PASCAL VOC) Segmentation
|Method|mIoU|ECE|
|---|---|---|
|DE-old|90.09|17.31|
|DE-new|86.96|12.36|
|NeuBoots|**90.14**|**6.00**|

2. ResNet-101 on 2D (PASCAL VOC) Segmentation
|Method|mIoU|ECE|
|---|---|---|
|DE-old|90.40|17.94|
|DE-new|87.48|11.52|
|NeuBoots|**90.56**|**6.14**|

3. U-ResNet on 3D (NIH3T3) Segmentation
|Method|mIoU|ECE|
|---|---|---|
|DE-old|59.71|1.78|
|DE-new|65.71|**0.94**|
|NeuBoots|**67.78**|1.67|

4. Scalable NAS on 3D (NIH3T3) Segmentation
|Method|mIoU|ECE|
|---|---|---|
|DE-old|60.64|1.39|
|DE-new|68.66|0.83|
|NeuBoots|**70.80**|**0.64**|

We will add these results in the updated version. Again, we deeply appreciate reviewers' scientific comments on the baseline models.

---

### Decision · Program_Chairs · 2021-09-27

**Decision:**

Accept (Poster)

**Comment:**

Reviewers recognized the importance of providing an efficient way to bootstrap neural nets for uncertainty quantification and praised the proposed approximate scheme.

During the review phase, reviewers highlighted how many experimental details and comparison w.r.t. modern neural uncertainty quantification methods were missing from the current version of the paper.
The authors provided an extensive rebuttal and managed to clarify the missing details and convince reviewers on certain points. However, some questions were left open: a predictive performance comparison is still missing as well as many references as highlighted by the reviewers.This work is accepted subject to the authors incorporating the promised changes and additional experimental results.